



# Comparative Study On Immersion Freezing Utilizing Single Droplet Levitation Methods

Miklós Szakáll[1], Michael Debertshäuser[1], Christian Philipp Lackner[1], Amelie Mayer[1], Oliver Eppers[2], Karoline Diehl[1], Alexander Theis[1], Subir Kumar Mitra[2], and Stephan Borrmann[1,2]

[1]Institute for Atmospheric Physics, Johannes Gutenberg University, Mainz; J.-J.-Becherweg 21, D-55128 Mainz; Germany
[2]Department of Particle Chemistry, Max Planck Instute of Chemistry; Hahn-Meitner-Weg 1, D-55128 Mainz, Germany

**Correspondence:** Miklós Szakáll (szakall@uni-mainz.de)

**Abstract.** Immersion freezing experiments were performed utilizing two distinct single-droplet levitation methods. In the Mainz vertical wind tunnel (M-WT) supercooled droplets of $700\,\mu m$ diameter were freely floated in a vertical air stream at constant temperatures ranging from $-5\,°C$ to $-30\,°C$ where heterogeneous freezing takes place. These investigations under isothermal conditions allow applying the stochastic approach to analyze and interpret the results in terms of the freezing or

nucleation rate. In the Mainz acoustic levitator (M-AL) 2 mm diameter drops were levitated while their temperature was continuously cooling from $+20°C$ to $-28\,°C$ by adapting to the ambient temperature. Therefore, in this case the singular approach was used for analysis. From the experiments, the densities of ice nucleating active sites (INAS) were obtained as function of temperature. The direct comparison of the results from two different instruments indicates a shift of the freezing temperatures towards lower values that was material dependent. As ice nucleating particles, seven materials were investigated, two represen-

tatives of biological species (fibrous and microcrystalline cellulose), four mineral dusts (feldspar, illite NX, montmorillonite, and kaolinite), and natural Sahara dust. Based on detailed analysis of our results we determined a material dependent temperature correction factor for each investigated particle type. The analysis allowed further classifying the investigated materials as single- or multiple-component. From our experiences during the present synergetic studies, we listed a number of suggestions for future experiments regarding cooling rates, determination of the drop temperature, purity of the water used to produce the

drops, and characterization of the ice nucleating material. The observed freezing temperature shift is significantly important not only for the intercomparison of ice nucleation instruments with different cooling rates but also for cloud model simulations with high speed ascents of air masses.

## 1  Introduction

Immersion freezing is considered to be the most effective freezing/nucleation process for ice particle production in mixed-phase

clouds (Diehl and Grützun, 2018). The nucleation abilities of atmospheric particles have been investigated very intensively for the last decades (Hoose and Möhler, 2012). Beside in-situ measurements, laboratory-based investigation techniques are widely used to discover the basic physical and chemical processes and properties of ice nucleating particles (INP). Laboratory immersion freezing experiments aim at the characterization of the temperature dependent ice nucleation ability of different



types of INP under controlled conditions. The ice nucleation efficiency of INP is commonly expressed in terms of ice nucleation
active sites (INAS) density $n_s(T)$. This gives the total number of nucleating sites per unit surface area of the particles that are
active between $0°C$ and the sub-zero temperature $T$ (DeMott, 1995; Connolly et al., 2009; Murray et al., 2012; Hoose and
Möhler, 2012). Another important parameter employed for describing the INP nucleation ability is the nucleation rate, i.e. the
probability of nucleation at a certain temperature per unit time per unit surface area of the particle (Vali, 2014).

Intercomparisons of measurement techniques revealed a wide scatter of measured ice nucleation activities of particles. This
is due to differences in the measuring methods employed by the different instruments and the diversity in the sample prepara-
tion at different research sites. One essential and still not understood discrepancy arises between dry dispersion and aqueous
suspension measurement techniques (Hiranuma et al., 2019). In the former, experiments employ water vapor condensation
onto dry dispersed particles followed by droplet freezing (e.g., cloud chambers, continuous-flow diffusion chambers), while
the latter denotes experiments starting with test samples pre-suspended in water before cooling (e.g., freezing arrays, drop lev-
itators). Several studies have focused on identifying potential reasons of this data diversity. Recently, two major international
research activities were conducted and produced a large amount of new data and results, one organized around the German
INUIT (Ice Nucleation Research Unit) research community, and the FIN (Fifth International Ice Nucleation Workshop). These
intercomparison campaigns revealed data diversity over several orders of magnitude in $n_s$ already among aqueous suspension
techniques also in case of a recommended protocol for sample treatment and preparation (Hiranuma et al., 2019; DeMott et al.,
2018). As these studies concluded, a key strategy would be to rigorously examine and define the functionality, configuration
and limitations of the measurement techniques and instruments (DeMott et al., 2018).

The most widely employed measurement instruments for investigating the immersion freezing of aqueous suspensions are
freezing arrays (Murray et al., 2011; Hader et al., 2014; Budke and Koop, 2015; Schrod et al., 2016; Reicher et al., 2018;
Harrison et al., 2018). Their inexpensive and easy operation, and the large number of simultaneously measurable droplets
offering good count statistics, made them standard devices for INP characterization experiments. However, the droplets are
continuously in contact with the supporting surface of the arrays which potentially introduces additional freezing sites and
initiates freezing. Furthermore, cross-contamination and evaporation or condensation during experiments might also influence
the measurement results (Budke and Koop, 2015).

To avoid the influence of the supporting surface, and to go a step further to real atmospheric conditions of cloud droplets,
single droplet levitation techniques are employed. They offer experiments with natural droplet shapes and contact-free levita-
tion, where the heat conduction of the released latent heat during freezing also better meets atmospheric conditions. The main
disadvantage of the single droplet levitation techniques is the limited number of individual droplet measurements they provide.
In order to get statistically relevant numbers of data points, a series of experiments has to be conducted by an operator over
a long time period, and, therefore, the long-term variation of the environmental conditions might lead to measurement uncer-
tainties. Prominent single droplet levitation techniques used for immersion freezing are an electrodynamic balance (Rzesanke
et al., 2012; Hiranuma et al., 2015) in which a charged supercooled droplet of about $100\,\mu m$ is levitated between electrodes; an
acoustic levitator (Ettner et al., 2004; Diehl et al., 2014), and a vertical wind tunnel (von Blohn et al., 2005; Diehl et al., 2011,





2014). An optical levitator for freezing experiments was also reported (Ishizaka et al., 2011), however, to our best knowledge it has not yet been applied for investigating immersion freezing.

In the Mainz vertical wind tunnel laboratory at the Johannes Gutenberg University of Mainz, Germany, we have conducted immersion freezing experiments with aqueous suspensions employing two independent single droplet levitation techniques. Our laboratory hosts two major facilities, both attaining contact-free levitation of liquid droplets and cooling of the surrounding air down to about $-28°C$. The main equipment is the Mainz vertical wind tunnel (M-WT) where atmospheric hydrometeors are freely suspended in an air updraft maintained by means of two vacuum pumps (Szakáll et al., 2010; Diehl et al., 2011). The

nature and the size of the hydrometeors suspended in the vertical air stream of M-WT are the same as in the real atmosphere. Furthermore, all hydrometeors are freely floated at their terminal falling velocities so that the relevant physical quantities, as for instance the Reynolds number and the ventilation coefficient (i.e. the ratio of the water vapour mass flux from the drop for the cases of a moving and a motionless drop), are equal to those in the real atmosphere. The instrumentation of the laboratory is complemented by a walk-in cold room in which the Mainz acoustic levitator (M-AL) is situated. In M-AL the free levitation

is achieved at the nodes of a standing acoustic wave (Ettner et al., 2004; Diehl et al., 2014).Although M-AL does not simulate atmospheric air flow conditions as M-WT, its simple setup, and the possibility of the direct measurement of drops' surface temperature promoted it for immersion freezing experiments (DeMott et al., 2018).

    The goal of the present study was to conduct a synergetic investigation of the immersion freezing ability of various INP using two qualitatively different free levitation methods. Furthermore, we aimed to provide direct intercomparisons of laboratory

instruments implementing different cooling rate conditions in immersion freezing experiments. Therefore, we carried out immersion freezing experiments in M-AL and M-WT by using aqueous samples of INP of different origin and types (biological particles, as well as proxy and natural mineral dusts). The theoretical background of the drop and INP characteristics in drop levitating techniques is summarized in Section 2. The experimental setups for the synergetic study employing M-WT and M-AL are introduced in Section 3. We present and discuss our experimental results in Section 4, and conclude with a summary

and an outlook for future experiments in Section 5.

## 2   Theoretical background of heterogeneous freezing

The heterogeneous nucleation of ice, i.e. the phase transition from liquid to solid state of water (Stage 2 in Fig. 1) induced by the growth of ice embryos on nucleation sites on INP, takes place at different temperatures, depending on the properties of the particles immersed in water (Vali, 2014). The larger the particle, the higher is the possibility that some part of its surface

favours ice nucleation. Hence, the probability of freezing (or nucleation) is dependent on the total surface area of the particles, or more specifically on the surface density of the ice nucleation active sites (Hoose and Möhler, 2012). Nevertheless, freezing is a dynamic process in which molecules from the metastable liquid state are joining to (and detaching from) the growing ice embryo. Therefore, nucleation is also a time-dependent process and occurs under isothermal conditions, as well (i.e. when the temperature remains constant). Specifically, the parametric description of immersion freezing and the interpretation of





the experimental results in the literature are based either on the stochastic (time-dependent) or on the singular (temperature-dependent) hypothesis, depending on the experimental conditions.

## 2.1 Stochastic approach

In experiments under isothermal conditions the number of unfrozen supercooled aqueous suspension droplets in a population decays exponentially with time, because at any point in time the number of freezing droplets is a function of the (decreasing)
number of still unfrozen droplets. The underlying assumption here is that each droplet freezes with the same probability when they contain identical INP. The rate $R_n$ which is used to describe this decay at a fixed temperature is determined from the number of the observed freezing events per unit time as (see Vali (2014) for detailed discussion)

$$R_n(t,T) = -\frac{1}{N_{tot} - n_{fr}} \frac{dn_{fr}}{dt} \tag{1}$$

where $n_{fr}$ denotes the number of frozen droplets at time $t$, and $N_{tot}$ the total number of droplets in the population, i.e. the total
number of the investigated individual droplets. After integrating Eq. (1) and assuming constant, i.e. time independent, cooling rate at a fixed temperature, the well-known expression of the freezing rate follows:

$$R(T) = -\frac{\ln\left(1 - \frac{n_{fr}}{N_{tot}}\right)}{t} \tag{2}$$

In the stochastic approach, the time dependence of nucleation is taken into account by introducing the heterogeneous nucleation rate coefficient $J_s$ – similarly to that for homogeneous nucleation (Pruppacher and Klett, 2010) –, which gives the rate of change
of the number of ice embryos per unit surface area of the ice nucleating particle. (In case of homogeneous nucleation $J_{hom}$ is given per unit volume of liquid drop.) If all droplets in the population contain the same amount of particle surface, and any part of the surface has an equal likelihood of containing an ice nucleating site (named single-component system after Broadley et al. (2012), and Herbert et al. (2014)), then by definition,

$$J_s(T) = \frac{R(T)}{S_p} \tag{3}$$

Here $S_p$ is the total particle surface area in each aqueous suspension droplet which can be calculated as,

$$S_p = V_d \cdot c \cdot SSA \tag{4}$$

with $V_d$ being the drop volume, $c$ the particle mass concentration in the sample solution, and $SSA$ the specific surface area of the particle. In case of any interparticle variability in the ice nucleating ability of the particle population Eq. 3 cannot be used. Such a system is called multiple-component, which, however, can be divided in subpopulations of equally ice active entities.
Each subpopulation $i$ can be treated as single-component and characterized by its number density $n_{s,i}$ and nucleation rate coefficient $J_{s,i}$ (Herbert et al., 2014).

## 2.2 Singular approach

The concept of the singular approach is based on the observation that freezing of drops containing INP occurs at a characteristic temperature once they are subjected to cooling. This also implies that supercooled droplets remain unfrozen arbitrarily long





when exposed to a Temperature $T'$, even if they contain INPs which however trigger freezing only at an INP-specific $T < T'$. Hence, the time dependence of ice nucleation is negligible in comparison to the particle to particle variability of the ice-nucleating ability (Connolly et al., 2009). Therefore, it is assumed that ice nucleation occurs on particular sites on the surface of a particle, the so-called ice-nucleating active sites (INAS), as soon as a temperature is reached which is characteristic for the INP material and its nucleating properties. Reaching this temperature by cooling, the droplet including the INP freezes

instantaneously. For the singular approach, the INAS surface density $n_S(T)$ is defined as the cumulative number of sites per surface area that become active between $0\,^{\circ}$C and $T$, and can be expressed as

$$n_s(T) = -\frac{\ln\left(1 - f_{ice}(T)\right)}{S_p} \tag{5}$$

where $f_{ice}(T) = \dfrac{n_{fr}(T)}{N_{tot}}$ is the so called frozen fraction, i.e. the cumulative fraction of droplets frozen between $0\,^{\circ}$C and $T$ in the population.

If all droplets were of same size and contained identical INP with homogeneous surfaces and uniform ice-nucleating sites, then $f_{ice}$ would be a unit step function at a characteristic temperature. Variability of INP in experiments arising from diverse composition, particle size, and location of INAS on particles' surface results in a distribution of characteristic temperatures, i.e. freezing probabilities of aqueous suspension droplets, which is represented by $f_{ice}(T)$ (Niedermeier et al., 2014).

### 2.3   Surface temperature of freely levitating droplets in freezing experiments

As becomes obvious from the description above, the correct representation of the drop temperature in freezing experiments is of crucial importance. In freezing array experiments the droplet temperatures are assumed to be equal to the substrate temperature which is directly measured by a thermometer. Since the contact area between a droplet and the substrate is large, this is an appropriate assumption even for relatively large drops with volumes of microliters. In single levitating techniques, as in M-WT or M-AL, the droplets are subjected to continuous cooling by heat diffusion and convection. The surrounding medium is air,

which is a far worse heat conductor than the substrates used in freezing array experiments. The effect of this adaptive droplet cooling becomes significant for drops with volumes in the microliter range (or, equivalently, with sizes in the millimeter range), because the amount of latent heat to be dissipated increases with volume as does the surface area of the drops.

The freezing process of a single aqueous solution droplet is depicted in Fig. 1. (The description hereinafter follows the concept of Hindmarsh et al. (2003).) First, the relatively warm droplet cools down after injection (Stage 1 in Fig. 1), and its

surface temperature $T_a$ approaches an equilibrium temperature $T_e$ determined by the ambient temperature $T_{inf}$, the dew point, and ventilation (Pruppacher and Klett (2010); and Appendix),

$$T_a(t \to \infty) = T_e \tag{6}$$

In case of an evaporating droplet the equilibrium surface temperature is always lower than the ambient temperature due to evaporative cooling. For a droplet in a continuous air flow, the temperature difference between the drop and its environment is

further enhanced by ventilation, resulting in a net temperature deviation $\delta$ (s. Appendix),

$$T_\infty - T_e = \delta \tag{7}$$

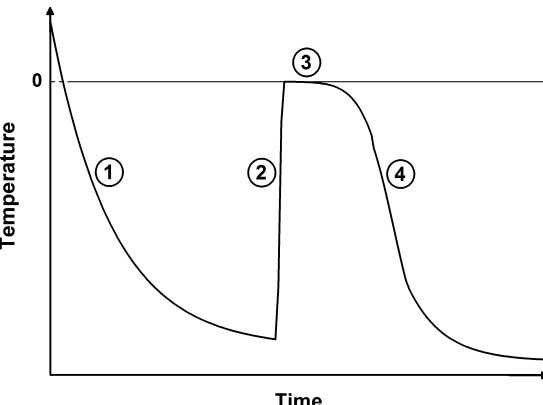

**Figure 1.** Schematic plot of the temporal surface temperature evolution of a freezing droplet. 1) Supercooling of the liquid droplet until nucleation is initiated; 2) Adiabatic freezing stage where rapid kinetic crystal growth takes place until supercooling is exhausted. No heat exchange with the environment; 3) Diabatic freezing stage in which ice crystal growth inside the droplet is governed by heat transfer with the environmental air; and 4) Cooling stage where the ice particle cools down adapting to the ambient temperature.

The temporal evolution of the surface temperature for a droplet placed in a cold environment is described mathematically by an exponential decay function (s. Appendix for the derivation):

$$T_e - T_a(t) = T_\infty - T_a(t) - \delta = [T_\infty - T_a(t=0) - \delta]\exp\left(-t/\tau\right) \tag{8}$$

with $\tau$ being the relaxation time, i.e. the time constant of the temperature adaptation. The main physical parameters that determine $\tau$, and therefore, the total cooling time of the droplet, are the drop size, the ventilation coefficient, as well as the ambient temperature and dew point (s. Appendix B2). Hence, for given experimental conditions, the temporal evolution of the drop's surface temperature in Stage 1 can be calculated using Eq. 8. In cold stage experiments freezing Stage 1 proceeds very quickly due to the large contact area (Harrison et al., 2018); in single levitation techniques this can take up to several

minutes. Drop freezing occurs at some instant in time or at some specific temperature. As soon as nucleation is initiated inside the supercooled drop rapid kinetic crystal growth takes place (Stage 2 in Fig. 1). This process is characterized by a sudden temperature increase due to the release of latent heat (which predominantly diffuses into the droplet) until the supercooling is exhausted, and the drop surface temperature rises to the ice-water equilibrium temperature (i.e. to $0\,°\mathrm{C}$ when the water activity of the investigated sample is $\approx 1$, as it was in our experiments). (For the drop freezing experiments this characteristic

temperature or time instant is to be measured, see Sections M-WT and M-AL.) Subsequently, a diabatic freezing of the whole droplet takes place (Stage 3). The temporal duration of this freezing stage is determined by the heat exchange between the hydrometeor and its environment, therefore it proceeds slower than Stage 2. In the end, the frozen particle cools down to the ambient temperature (Stage 4).




In practice, $n_s$ can be determined when a population of aqueous suspension droplets is continuously or stepwise cooled down, and the number of freezing events as the function of temperature is registered. Cooling rates in freezing array (or cold stage) experiments range from 1 to $10\,\mathrm{K min}^{-1}$, representing also typical atmospheric rates. Employing a constant cooling rate $r$ in the experiments, the temperature decreases with $\delta T = r\delta t$ within a time interval $\delta t$. The number of droplets freezing in this time interval at temperature $T$ can be calculated using the multiple-component system approach of Herbert et al. (2014) as

$$\delta N_{fr} = \sum_{i=1}^{n} n_{liq}\left(1 - \exp\left(-J_{s,i}(T) \cdot A_i \cdot \delta t\right)\right) \tag{9}$$

where $i$ denotes the components in the system, $A_i$ is the surface area of component $i$, and $n_{liq,i}$ is the number of unfrozen droplets at the beginning of the time interval. Hence, employing a higher cooling rate in the experiments results in a decrease in $\delta t$ for a fixed $\delta T$, which, in turn, decreases the probability of a freezing event to occur between $T$ and $T - \delta T$. Thereby the cumulative frozen fraction spectrum and, consequently, $n_s(T)$ will be shifted to lower temperatures. This temperature shift for any given change in cooling rate from $r_1$ to $r_2$ can be calculated following Herbert et al. (2014) as:

$$\Delta T_f = \frac{1}{\lambda} \ln\left(\frac{r_1}{r_2}\right) \tag{10}$$

The parameter $lambda$ is the gradient of the logarithm of the nucleation rate coefficient in a simple temperature-dependent formulation of $J_s$ introduced in recent immersion freezing studies (Murray et al., 2011; Broadley et al., 2012; Wright and Petters, 2013; Herbert et al., 2014):

$$\ln J_s(T) = -\lambda \cdot T + \phi \tag{11}$$

In Eq. (11) $\phi$ is the relative nucleating efficiency of the INP (see Herbert et al.,2014). It is obvious from Eq. (10) that experiments employing different cooling rates provide different INAS spectra due to the difference in their related temperature shifts $\Delta T_f$. In order to become independent of this systematic artefact, the experimental cooling rate $r_{exp}$ is normalized to a standard value, typically to $1\,\mathrm{K\,min}^{-1}$, which results in

$$\Delta T_f = \frac{1}{\lambda} \ln\left(\frac{1}{|r_{exp}|}\right) \tag{12}$$

Similarly, in isothermal experiments the relative change in cooling rate can be expressed as a relative change in residence time, i.e. the period of time for which the particles are exposed to a constant temperature (Herbert et al., 2014). In these experiments the temperature shift can be calculated as,

$$\Delta T_{iso} = \frac{1}{\lambda} \ln\left(\frac{\lambda \cdot t_{res}}{t_{std}}\right) \tag{13}$$

where $t_{res}$ is the characteristic residence time of droplets under isothermal condition in the experiments, and $t_{std}$ is a standard time required for an isothermal experiment to be comparable to a normalized cooling experiment (i.e. $t_{std} = 60\mathrm{s}$).

For comparative analysis of the ice nucleating ability of particles investigated by different experimental approaches, $\lambda$ is a crucial parameter. Large values of $\lambda$ indicate effective INP and, therefore, weak time dependence, while less effective INP





possess small $\lambda$ values. Herbert et al. (2014) determined $\lambda$ for a set of ice nucleating materials and compared to several literature
data. The large variability of $\lambda$ on the material of the INP necessitates further quantification of $\lambda$ for other atmospherically
relevant INP species. This is discussed in Section 4 in the context of our experiments.

Another important quantity which is used for discussing the results of the present experiments is the temperature derivative
of the logarithm of the freezing rate normalized by the aerosol total surface area (Vali, 2014):

$$\omega = -\frac{d\ln(R/A)}{dT} \tag{14}$$

Note that for a single-component system Eq. (3) can be applied (i.e. $R/A = J_s$), and therefore, $\omega = \lambda$ , while for a multiple-
component system $\omega \neq \lambda$ (Herbert et al., 2014).

In this study we measured the frozen fraction $f_{ice}(T)$ of different INP with two different methods. From $f_{ice}(T)$ the INAS
densities $n_s(T)$ and the freezing rates $R(T)$ were determined. Based on these quantities the temperature shifts $\Delta T_f$ and $\Delta T_{iso}$,
and the material parameters $\omega$ and $\lambda$ were calculated and analyzed.

## 3 Methods

### 3.1 Material and sample preparation

The experiments were carried out using seven different types of materials which are listed in Table 1. All of these materials are
considered to be important constituents of atmospheric ice nucleation particles. As biological INP surrogates we investigated
two cellulose types, microcrystalline and fibrous cellulose (hereafter MCC and FC, respectively). Among the investigated
mineral dust materials feldspar (especially K-feldspar) exhibits the highest ability to initiate ice formation. It is a prevalent
component of desert dusts so that by scaling down it is representative for dust samples in dependence on their composition
(Atkinson et al., 2013). Illite NX can be considered as proxy for desert dust since their mineralogical compositions are simi-
lar (Broadley et al., 2012). Montmorillonite K10 and kaolinite (SigmaAldrich) are commercially available and characterized
mineral dust materials of relevance for the atmosphere, which also have been subject of several previous studies. Further-
more, we used a natural desert dust particle sample, the ice nucleation abilities of which have been investigated with different
measurement techniques during the INUIT09 measurement campaign (Ullrich et al., 2019).

Atmospherically relevant INP exhibit an extremely wide range in their ability to heterogeneous freezing. Furthermore,
there is a large spread in the specific surface area (SSA) of the investigated materials from around $1\,\mathrm{m^2\,g^{-1}}$ to $245\,\mathrm{m^2\,g^{-1}}$ (s.
Table 1). We chose therefore diverse mass concentrations for each of the different particle types to obtain reasonable numbers of
freezing events within the temperature ranges of our measurement facilities. Furthermore, since the volume of the investigated
droplets in M-AL was approximately 20 times larger than in M-WT, we used reduced mass concentrations in M-WT to obtain
overlapping freezing curves with the two methods (s. Eq. (5)).

Prior to each set of experiments, 20 to $40\,\mathrm{mL}$ aqueous suspension was prepared by mixing sample particles of known weight
(measured by an analytical balance from Sartorius) with high purity water (CHROMASOLV water for HPLC, Sigma-Aldrich).
Between the measurement runs the aqueous suspension was continuously stirred at a very low rate using a magnetic stirrer to





**Table 1.** Aerosol material and sources measured in the current study. Also given are the specific surface area (SSA), and the concentrations used for the immersion freezing experiments in M-AL and in M-WT.

| Sample material | SSA $(m^2\,g^{-1})$ | Concentration $(g\,L^{-1})$ |
|---|---|---|
| Fibrous cellulose[a] (FC; Sigma, C6288) | $1.31 \pm 0.1$ | 1.0 |
| Microcristalline cellulose[a] (MCC; Aldrich, 435236) | $1.44 \pm 0.1$ | 1.0 |
| Feldspar[b] (Microcline) (IAG TU Darmstadt) | 1.79 | 0.5 / 0.66 / 0.8 |
| Illite NX[c] (Arginotec) | $124.4 \pm 1.5$ | 0.25/2.5 |
| Kaolinite (Sigma Aldrich) | 8.33 | 0.1/1.0/1.265 |
| Montmorrillonite K10[d] (Sigma-Aldrich) | $245 \pm 20$ | 5.0 |
| SDB01[0] (Bodele-Depression, Ts) | 26 | 1.0 / 0.1 |

[a] Same as used in Hiranuma et al. (2018)
[b] Same as FS01 in Peckhaus et al. (2016)
[c] Same as in Diehl et al. (2014) and Hiranuma et al. (2015)
[d] Same as in Diehl et al. (2014)
[d] Same as in Ullrich et al., (2019)

avoid coagulation and sedimentation of the particles in the suspension. A hypodermic syringe was used to inject suspension droplets into the measuring instruments. For the M-AL measurements, the syringe was filled with aqueous suspension after an idle time of about 30 minutes without stirring (following the sample preparation protocol of Hiranuma et al. (2019)), so that at the uppermost part of the solution a homogeneous suspension was generated. For the M-WT measurements we abandoned an idle time, because in this case we could presume already homogeneous suspension due to the low particle concentration. Furthermore, the syringe was shaken prior to droplet injection both in M-AL and M-WT experiments to ensure homogeneous particle distribution in droplets. Otherwise no pre-treatment procedures were applied.





**Table 2.** Characteristics of the experiments conducted with M-WT and M-AL.

| Characteristics | M-WT | M-AL |
| --- | --- | --- |
| Thermal condition | isothermal | continuous cooling |
| Droplet cooling time | 4 to 6 s | 10 to 120 s |
| Freezing approach | stochastic or singular | singular |
| Deliverables | $n_s$, $R/A$ | $n_s$ |
| Temperature range | $-10$ to $-30\,°$C | $-15$ to $-25\,°$C |
| Droplet diameter | $700\,\mu$m | $2\,$mm |
| volume | $0.18\,\mu$L | $4\,\mu$L |

## 3.2 Experimental setups and procedures

The characteristics and deliverables of the M-WT and M-AL instruments essential for the present study are summarized in
Table 2 and will be described in the following subsections. Detailed descriptions of the experimental facilities are given, e.g.,
in (Szakáll et al., 2010; Diehl et al., 2011, 2014).

### 3.2.1 M-WT (Mainz vertical Wind Tunnel)

The Mainz vertical wind tunnel (M-WT) is a world-wide unique experimental facility designated for the laboratory investiga-
tion of atmospheric hydrometeors, such as cloud droplets, raindrops, graupel, hailstones, and snowflakes. Single hydrometeors
are floated freely at their terminal velocities in the laminar vertical updraft of the wind tunnel. Hence, the relevant physical
properties of the hydrometeors, such as Reynolds numbers and ventilation coefficients, are equal to their values in the real
atmosphere (Szakáll et al., 2010; Diehl et al., 2011).

For the immersion freezing experiments the air in the M-WT was cooled down and kept constant (within $\pm 0.3\,°$C) at various
temperatures between $-15$ and $-30\,°$C. Thus, the experiments presented here were conducted under isothermal conditions.
For appropriate measurement statistics, at each temperature, particle type, and INP concentration, a total number of 70 aqueous
suspension droplets were investigated. After injection, each droplet was floated in the M-WT until it froze, or until the experi-
ment was terminated because of reaching a predefined time limit. The onset of freezing is characterized as a sudden significant
change in floating behavior of the droplet caused by the irregular shape of the frozen particle. This changing behavior was
visually observed and registered by the operator during the experiments. In this way, the total observation time, i.e. the time
duration from injection until the onset of freezing was recorded (Diehl et al., 2014). When a droplet did not freeze within
35 seconds, it was counted as unfrozen. In our earlier immersion freezing studies we levitated the supercooled droplets for
at most 30 seconds (if freezing was not initiated sooner). We extended this total observation time with 5 seconds to consider
the approximate time period a drop needed to approach its equilibrium temperature in M-WT (Fig. 2). Furthermore, wind





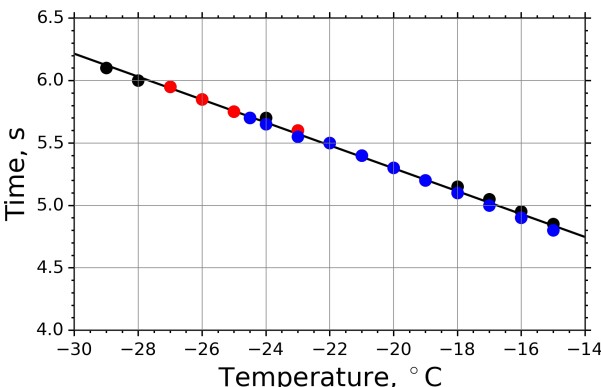

**Figure 2.** Time needed to approach the equilibrium temperature $T_e$ with an accuracy of $0.2\,°C$ for dew point temperatures $-30$, $-27.5$, and $-25\,°C$ (plotted by black, red, and blue dots, respectively). The data points were calculated using Eq. (B21). The regression line is $t_a = 3.46 - 0.09T$ ($T$ in $°C$).

speed, air temperature, and dew point temperature (typically in a range from $-20$ to $-35\,°C$) in the wind tunnel were recorded

continuously at $2\,Hz$ temporal resolution.

The drop surface temperature was calculated using Eq. (B21) considering thermal steady state condition between the levitating drop and its surrounding air. The time needed to approach the equilibrium temperature in the M-WT experiments within $0.2\,K$ difference (i.e. below the temperature measurement precision of the applied PT100 sensor) was calculated at distinct M-WT air temperatures and plotted in Fig. 2 for different dew points. In the calculations the starting drop temperature was set

to $20\,°C$, and the time period for reaching a temperature difference of $0.2\,K$ (i.e. below the temperature measurement precision of the applied PT100 sensor) between the drop and the equilibrium temperature was computed. One can observe a slight dependence of the adaptation time on the air temperature, but $T_e$ was typically reached within 5 to 6 seconds for air temperatures between $-15$ and $-28\,°C$. In the calculations, dew points of $-30$, $-27.5$, and $-25\,°C$ were applied and the results are merged in the plot shown in Fig. 2. Hence, the adaptation time was found to be practically independent on the dew point for the M-WT

experiments.

Typical droplet diameters/volumes were approximately $700\,\mu m/0.18\,\mu L$. The size of each investigated droplet was determined from its terminal velocity (Beard, 1976), i.e. from the vertical air speed needed for freely suspending it, which can be measured with high accuracy in the M-WT (Diehl et al., 2014).

Since the M-WT experiments represent isothermal measurement conditions, the stochastic approach was applied first for

data analysis. Thus, the rate constant $R$ and the nucleation rate coefficient $J_{het}$ were calculated from Eqs. 2 and 3, respectively, using the number of freezing events as function of the freezing time. In the analysis the freezing time of each droplet was calculated by subtracting the adaptation time (Fig. 2) from the total observation time lasting from droplet injection until the onset of freezing.



Furthermore, from the number of freezing events over the whole observation time period the frozen fraction $f_{ice}(T)$, and the INAS density $n_s(T)$ (Eq. 5) were determined employing the singular approach.

Background measurements were carried out before each experimental run by floating at least 10 HPLC (high purity liquid chromatography) water droplets for 35 seconds in the tunnel. We have not observed any freezing event during these test measurements, which indicate the absence of impurities (i.e. background active INP) both in the HPLC water droplets and in the wind tunnel.

### 3.2.2   M-AL (Mainz Acoustic Levitator)

The main component of the M-AL measurement facility is an acoustic levitator (APOS BA 10, tec5 GmbH), in which contact-free single droplet levitation is maintained by a standing ultrasonic wave (Diehl et al., 2014). The M-AL is placed inside a walk-in cold room where the ambient temperature was set to be $-30\,°C$ for the freezing experiments. In order to prevent any disturbing air motion, which might cause unsteady temperature condition and unstable levitation, or carry ice nucleating particles onto the levitating drop surface, the M-AL was surrounded by a protective acrylic housing. Using this setup, the air temperature in the M-AL was $-28\,°C$ as measured by a PT100 sensor. An infrared thermometer (KT 19.82 II, Heitronics) and a digital video camera (USB-CAM-103H, Phytek GmbH) were arranged around the acrylic housing of the levitator.

One of the main advantages of the experimental setup of M-AL is the direct observation of the surface temperature of the levitated drops during the cooling-freezing process, which was performed by the infrared thermometer at a rate of $2\,Hz$. The minimum observable spot size of the infrared thermometer restricted the minimum levitated drop diameter to $2\,mm$. The actual drop size was determined from the images captured by the digital video camera instantaneously after injecting the drop into M-AL. An example of a video recorded during an experiment on the ice nucleation ability of cellulose is provided as Video supplement of this paper (see https://doi.org/10.5446/46729). In the video the air temperature in the cold room measured by a PT-100 sensor, the continuously determined drop size (as the volume equivalent diameter), and the drop surface temperature measured by the infrared thermometer are displayed. The recorded drop cools continuously adapting its temperature to the ambient temperature until the freezing is initiated at about $-21.8\,°C$. The onset of freezing can be observed by the sudden change in the transparency of the droplet, and the increase of the drop surface temperature to $0\,°C$.

In case of M-AL experiments, the droplet surface temperature approached to the equilibrium temperature in a slower manner as in M-WT, which was primarily due to the larger drop size and smaller ventilation effect stemming from the acoustic field (s. Appendix B3). The relatively moderate cooling and large drop surface area enabled us to determine the freezing temperature of the individual drops with high accuracy by the infrared thermometer. In Fig. 3 two typical examples of M-AL measurements are plotted: In one case (black line) no freezing occurred and the experiment was terminated after $80$ seconds measurement time. In the other case (red line), freezing was initiated after 35.5 seconds cooling at about $-21.3\,°C$ surface temperature. The arithmetic mean of the three recorded temperatures preceding the deepest drop surface temperature during the last $1.5$ seconds before the onset of freezing, i.e. the temperature at transition from Stage 1 to Stage 2 in Fig. 1, was considered as the freezing temperature. The number of frozen droplets was measured and binned in $1\,K$ intervals to calculate $f_{ice}$, and thereof $n_s$ applying the singular approach (Eq. 5).





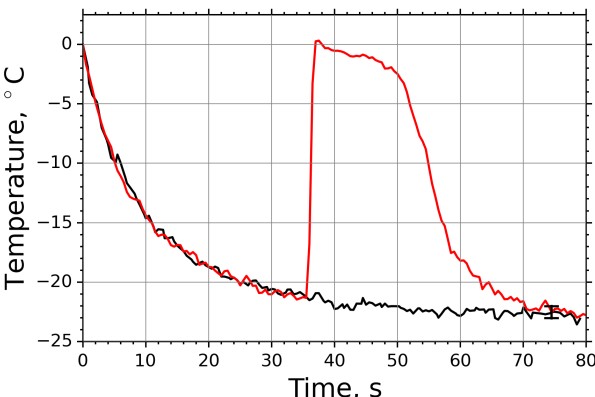

**Figure 3.** Measured surface temperatures of droplets levitated in M-AL: examples of freezing (red line) and non-freezing (black line) events. The measurement uncertainty of the temperature was $\pm 0.5\,\mathrm{K}$.

The temporal evolution of the drop temperature in the sample experimental run of M-AL depicted in Fig. 3 can be described by Eq. (B16) with $\tau = 11.3\,\mathrm{s}$, applying a ventilation coefficient of $5.6$ which is in accord with the findings of Lierke (1995). The relaxation time $\tau$ was determined for each experimental run in M-AL and showed typical values between $8.94\,\mathrm{s}$ and $15.42\,\mathrm{s}$.

The actual cooling rate at a time instant during temperature adaptation is defined as $r(t) = -dT/dt$, that can be calculated after rearranging Eq. (8) to

$$T_a = T_e - (T_e - T_a(t=0))\exp(-t/\tau) \tag{15}$$

After some manipulation the actual cooling rate can be written as,

$$r(T) = \frac{T - T_e}{\tau} \tag{16}$$

The $r(T_a)$ curve for $\tau = 11.3\,\mathrm{s}$ is shown in Fig. 4. It is apparent from the figure that at high temperatures the cooling rate is substantially high and gets moderate values only at low temperatures close to the equilibrium temperature. For such large cooling rates in M-AL measurements Eq. 12 predicts significant temperature shift in drop freezing temperature.

## 4 Results and Discussion

In this section we present the results of M-WT and M-AL experiments on immersion freezing using the clay mineral kaolinite. The data for other materials listed in Table 1 are presented in the Supplement Material.

### 4.1 M-WT experimental results

INAS densities computed from cumulative nucleus spectra obtained from M-WT measurements of kaolinite with concentrations of $0.1\,\mathrm{g\,L^{-1}}$ and $1.0\,\mathrm{g\,L^{-1}}$, and marked with light and dark blue symbols, respectively. The number of data points is



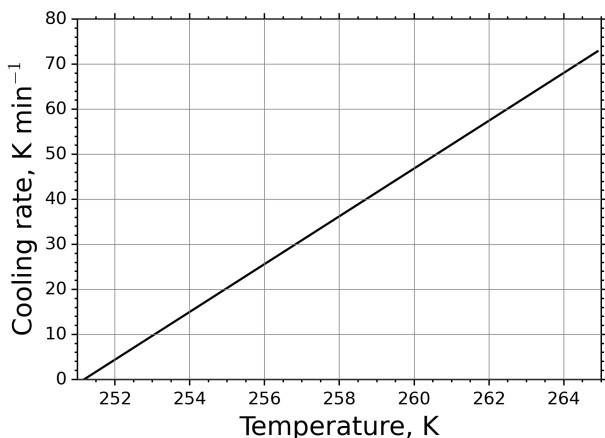

**Figure 4.** Actual cooling rate as the function of temperature in M-AL for the example shown in Fig. 3, with $\tau = 11.3\,\mathrm{s}$.

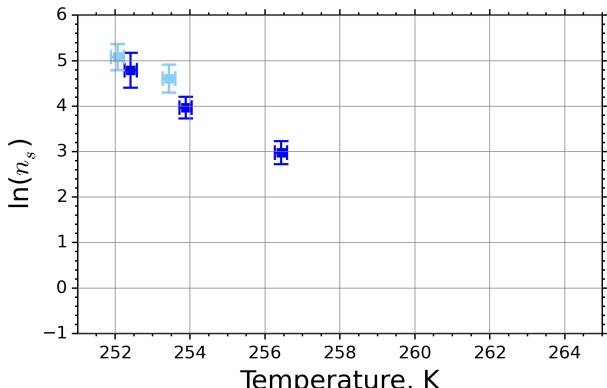

**Figure 5.** INAS density for kaolinite as function of temperature determined from the frozen fraction of $0.1\,\mathrm{g\,L^{-1}}$ and $1.0\,\mathrm{g\,L^{-1}}$ suspension drops (marked with light and dark blue, respectively) investigated in M-WT. Each data point represents 70 individually measured droplets, each of which with diameter of approximately $700\,\mathrm{\mu m}$. The error bars are representing the $1\sigma$ values of the measured air temperatures and the calculated drop sizes.

limited to five which is the issue of the M-WT experiments being very laborious for collecting statistically relevant numbers of measurements for each temperature. Comparing Fig. 5 with the INAS densities of other investigated materials presented in the Supplement reveals that kaolinite is a good atmospheric INP exhibiting large ns values that, nevertheless, vary steeply over one order of magnitude within the investigated temperature range of only $4\,\mathrm{K}$, i.e. here from $252\,\mathrm{K}$ to $256\,\mathrm{K}$.





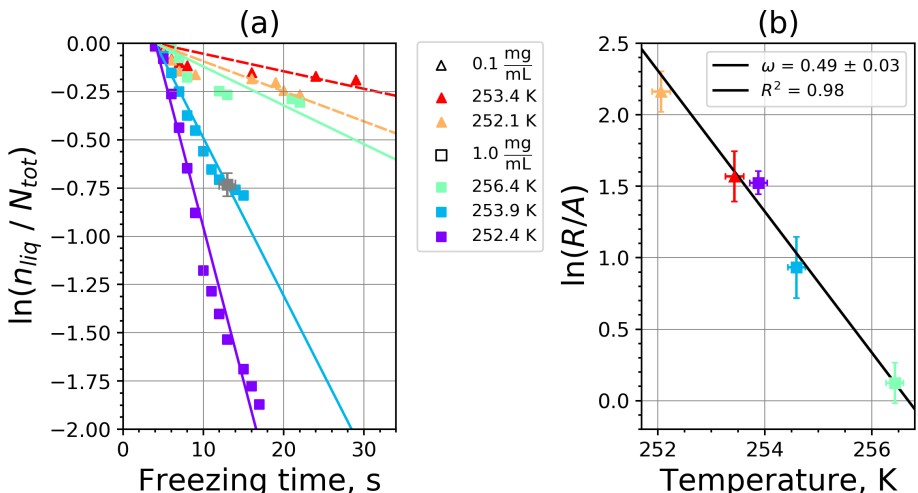

**Figure 6.** Kaolinite: (a) The decrease of fraction of droplets which remained liquid with time at different temperatures in the isothermal experiments in the M-WT. The colors are corresponding to different temperatures; experiments with particle concentrations of $0.1\,\mathrm{g\,L^{-1}}$ and $1\,\mathrm{g\,L^{-1}}$ are plotted by triangles and rectangles, respectively. The grey symbol marks a data point for $1\,\mathrm{g\,L^{-1}}$ at $253.9\,\mathrm{K}$ and indicates typical error bars. (b) Freezing rate of kaolinite normalized to surface area as function of temperature calculated from (a) using Eq. (17). The horizontal error bars are the $1\sigma$ values of the measured temperatures, while the vertical error bars are representing the fit error in R/A calculation.

For computing $n_s$ for Fig. 5 using Eq. (5), the fraction of frozen droplets $f_{ice}$ was determined employing the singular

approach, i.e. by counting the number of droplets frozen in an experimental run disregarding the time from injection until freezing. A droplet remaining liquid for up to 35 seconds (i.e. the end of the experimental run) was classified as unfrozen.

From the time resolved measurement data from M-WT, the time dependence of the freezing process was analysed. For that, Eq. (2) was rearranged to

$$\ln\left(\frac{n_{liq}(T,t)}{N_{tot}}\right) = -R(T)\cdot t \tag{17}$$

where $n_{liq}(T,t) = 1 - n_{fr}(T,t)$ is the number of droplets remaining liquid after time $t$ at temperature $T$. Fig. 6 depicts the time dependence of liquid ratio from the M-WT measurements at five different temperatures and using the two distinct concentrations of kaolinite as for Fig. 5. The times needed for the injected droplets to reach their equilibrium temperatures (i.e. 6 seconds, see Fig. 2) were subtracted from the recorded time interval between injection and freezing. At lower temperatures and with higher particle surface areas per drop (i.e. higher INP concentration), the curves are getting steeper indicating that freezing

proceeds faster. Fig. 6a clearly shows the expected exponential decay of liquid drops with time confirming the application of the stochastic approach for the M-WT experiments. The temperature dependence of the normalized freezing rate according to Eq. (17) as shown in Fig. 6b was determined by computing the slopes of the curves in Fig. 6a and dividing them by the total surface areas of INP immersed in the examined water droplets. Fig. 6b reveals the expected linear dependency of $R/A$





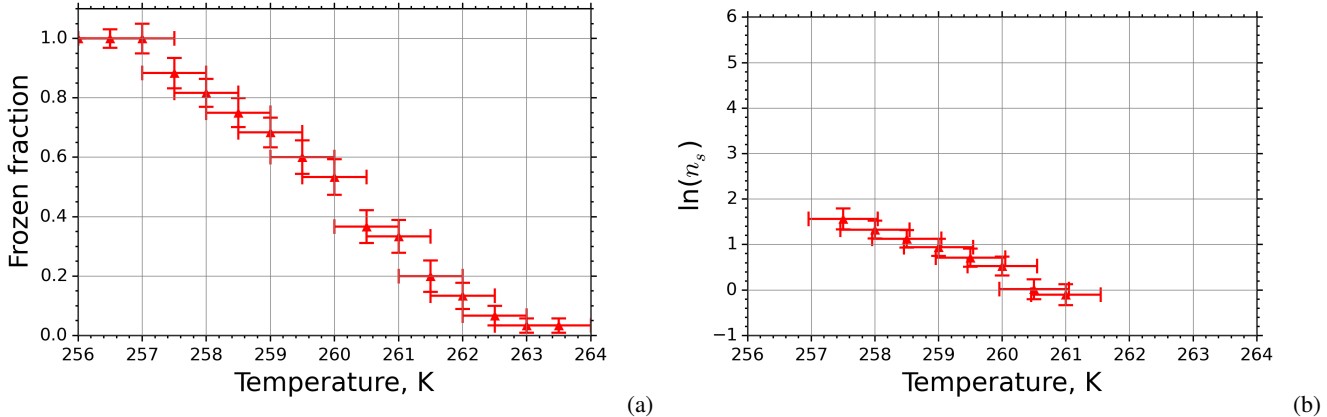

(a)

(b)

**Figure 7.** (a) Frozen fraction of $2\,\mathrm{mm}$ aqueous suspension droplets containing $1.265\,\mathrm{g\,L^{-1}}$ kaolinite measured in M-AL. (b) INAS density of kaolinite as function of temperature determined from the spectrum shown in (a).

in agreement with Eq. 14. Hence, $\omega$, which is the slope of $R/A$ (see Eq. (14)), can readily be determined for the investigated

kaolinite sample from Fig. 6b by linear regression as $\omega = 0.49 \pm 0.03$. Note that if the INP can be considered as single-component, then $J_s = R/S_p$.

## 4.2   M-AL experimental results

Frozen fractions of kaolinite suspension with $1.265\,\mathrm{g\,L^{-1}}$ concentration as function of temperature are shown in Fig. 7a. Error bars are associated to the temperature bin interval ($\pm 0.5\,\mathrm{K}$), and the uncertainty in the determination of $f_{ice}$ stems from the

counting statistics and the experimental temperature uncertainty. The active site density $n_s$ calculated from Eq. 5 using $f_{ice}$ in Fig. 7a is plotted in Fig. 7b. Here the error bars originate from Gaussian error propagation when using the measured data in Fig. 7a. From the calculation we excluded the data points for which $f_{ice}$ was above 90% or below 10%. This cut-off was introduced because in these cases the uncertainty of $f_{ice}$ was very large due to the poor counting statistics when freezing or unfreezing events occur very rarely.

Another criterion to use $f_{ice}$ for further evaluations was that it should significantly exceed the background caused by impurities in the water used for generating the aqueous suspension. To determine this background spectrum, we investigated pure water droplets before each experimental run in M-AL, similarly to the M-WT measurements. However, in contrast to the findings for M-WT, some of the HPLC water droplets froze in M-AL. This indicates that the abundance of impurities in the HPLC water was high enough in the relatively large ($\sim 4\,\mathrm{\mu L}$) drops in M-AL to initiate freezing. Therefore, cumulative nucleus

spectra for pure water samples were calculated similarly as for the INP (Fig. 8). Although the number of freezing droplets was relatively small at temperatures higher than $248\,\mathrm{K}$, in some cases (e.g., when low concentrations were used) the INP nucleus spectra had to be corrected by considering the water background spectrum as described below.





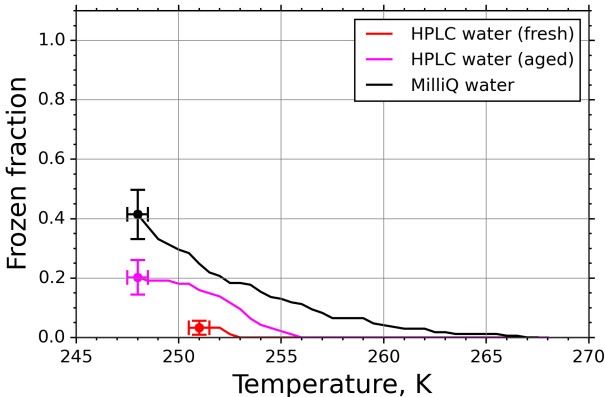

**Figure 8.** Freezing spectra of water of different purity grades: High precision liquid chromatography water (HPLC, SigmaAldrich), and in-house purified MilliQ water.

In earlier experiments in M-AL (e.g., Diehl et al. (2014)), in-house produced MilliQ water was used as solvent for the aqueous solutions. Therefore, we also analyzed the MilliQ water in our present experiments. The results are plotted by black

symbols in Fig. 8. Apparently, using HPLC water from a freshly opened chemical bottle (red symbols) reduces the background $f_{ice}$. Nevertheless, the $f_{ice}$ spectrum of HPLC water changed with time and increased significantly after about a year (magenta symbols), indicating an aging effect. This behavior of different water types is in accord with the finding of Hiranuma et al. (2019). Since it is difficult to eliminate the contribution of INP still present in high purity water (see Fig. 8, and Whale et al. (2015)), we applied a background correction method described below using the $f_{ice}$ spectrum for pure water drops collected

prior and during each set of experiments.

The background spectra were also corrected by shifting the freezing temperature following Vali (2014) with

$$\beta = 0.66 \lg(r) \tag{18}$$

Although Vali proposed the factor $0.66$ for the temperature correction of pure water, this parameter depends most probably on the type of impurities in the water. This is also suggested by Fig. 8 since the frozen fraction spectra are significantly different

for different water samples, purity grades, and water age. Nevertheless, the temperature correction of Eq. 18 barely shifts the background spectra: $\beta = 0.46$ for $252\,\mathrm{K}$ where the cooling rate in M-AL is approximately $252\,\mathrm{K}\,\mathrm{min}^{-1}$. Such a temperature shift would increase the background frozen fraction by less than $0.05$. Therefore, no background subtraction (as, e.g., in Hader et al. (2014)) was applied but a cut-off temperature was defined where the difference between the background and the INP spectra was less than $0.05$. This correction method was only necessary for FC and MCC in our experiments, while the other

materials initiated freezing at higher temperatures for the investigated concentrations.





## 5 Reconciling the M-WT and M-AL experimental data by temperature correction

Plotting the INAS densities obtained from M-WT and M-AL experiments, respectively, in one figure reveals an apparent shift of the curves either in $\ln(n_s)$ or in the temperature (Fig. 9a). This shift was found for all investigated materials but with different magnitudes (s. Appendix A). Curves of INAS as function of temperature from the same experimental methods (M-AL or M-WT) but measuring different INP concentrations (s. Table 1) do not spread in such a systematic way, which indicates that the shift stems very likely from the detected freezing temperatures. Since M-AL exhibits a very large cooling rate for temperatures higher than $255\,\mathrm{K}$ (see Fig. 4), a temperature shift predicted by Eq. 12 can be significant for some given materials depending on their $\lambda$ values. Nevertheless, we thoroughly checked other possible sources of any systematic freezing temperature shift. One obvious issue might arise from the relatively large volume of the drops examined in M-AL. In the experiments the surface temperature of the drops was continuously measured, however, if the drop cools down at a high rate, heat from the drop interior might not be transported outward sufficiently quickly. Some INP are located inside the drop, i.e. away from the drop surface, hence, they would experience higher temperatures than measured by the IR thermometer. This might falsify the experimentally determined temperature dependence of the ice nucleating ability. Nevertheless, our computation on the temporal evolution of a continuously cooling drop showed a temperature difference of maximum $0.5\,\mathrm{K}$ between the drop interior and surface which is within the measurement error of M-AL (s. Appendix B4). This temperature difference is higher at higher temperatures, where fewer freezing measurements were carried out. At surface temperatures below $258\,\mathrm{K}$ the difference is about $0.2\,\mathrm{K}$ only. Furthermore, the number of kaolinite particles in a $0.1\,\mathrm{g\,L^{-1}}$ aqueous suspension drop of $2\,\mathrm{mm}$ diameter, for instance, is approximately 300000. Thus, numerous particles will occur in the coldest region of the drop. Since a single particle is sufficient to initiate nucleation, the warmer temperature in the drop interior plays a minor role in initiating the freezing.

To correct the measurement data for a temperature shift due to interparticle variability of ice nucleation efficiency, we follow the approach of Herbert et al. (2014) as described in Section 2. We present here only the case of kaolinite as an example; the approach was applied to reconcile the data for all examined materials. Those results are presented in Appendix A. The procedure for correcting for the raw data set in M-AL and M-WT is depicted as a flow diagram in Fig. 10.

### 5.1 Determination of $\lambda$

The optimal parameter $\lambda$ for the temperature shift was determined assuming that with the correct $\lambda$ value the $\ln(n_s)$ data from the two experiments M-WT and M-AL converge onto one single curve. Therefore, the temperatures of the uncorrected data were modified by applying Eq. (13) on the isothermal experimental data from M-WT, and Eq. (12) on the data obtained using continuous cooling approach of M-AL. For each investigated INP species a set of $\lambda$ values varying from 0.1 to 8.0 with 0.1 applied for the modification in Eqs. (13) and (12). A linear fit to the derived $\ln(n_s(T))$ (black solid line and black data points in Fig. 11a) and the RMSE (root mean square error) between the data and the linear fit were calculated for each set of modified experimental data. The RMSE for the set of $\lambda$ values for the kaolinite experiments is depicted in Fig. 11b. The optimal $\lambda$ value, 1.7 in the present case, is corresponding to the minimum of the RMSE curve. This optimal value provides the best linear fit among the tested $\lambda$s.





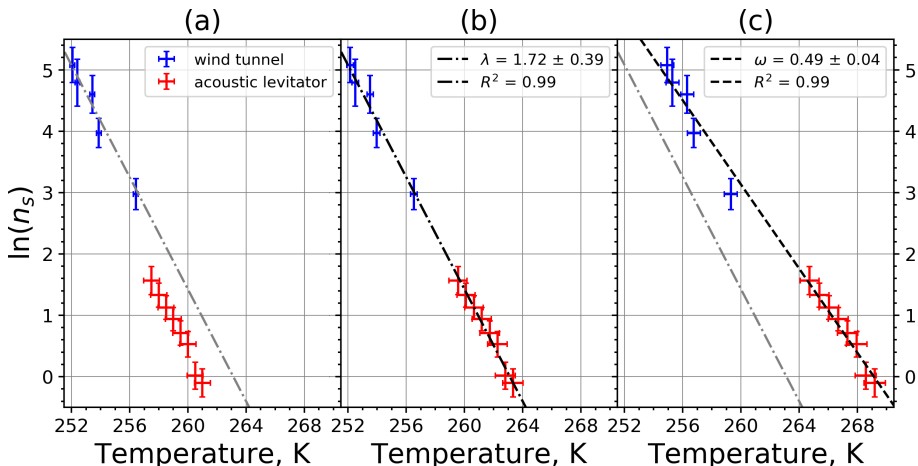

**Figure 9.** (a) Composite INAS density spectrum of kaolinite from the uncorrected M-WT (blue) and M-AL (red) measurements. (b) and (c) show the temperature-corrected data points from the M-WT and M-AL experiments based on on $\lambda$ and $\omega$, respectively. The dashed-dotted line in (c) is the regression line for the corrected data points obtained by employing the optimal $\lambda$-value as in (b).

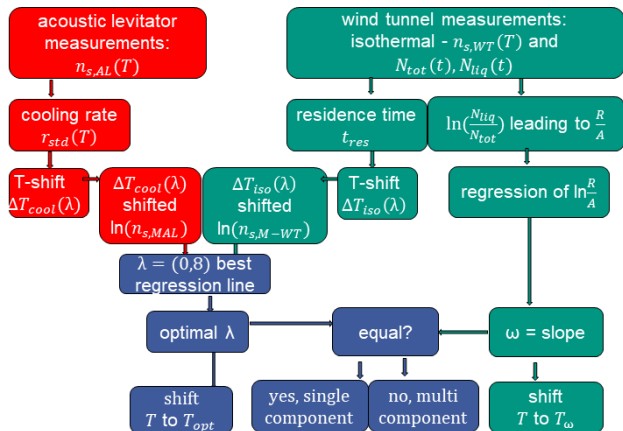

**Figure 10.** Flow chart of the procedure used to combine M-WT and M-AL measurement data for determining $\lambda$ and $\omega$ and classifying the investigated materials as single or multiple-component.

To determine the error of $\lambda$ originating from the measurement error, the following procedure was used. We generated random
data points around each of the actually measured data points, but within the bounds of the measurement error (assuming the error bar of the measurement corresponds to 95% confidence interval). Hence, the number of data points for the $\lambda$-analysis did not change but each data point was shifted both in temperature and in $n_s$. Then, the optimal $\lambda$ value for this modified data set



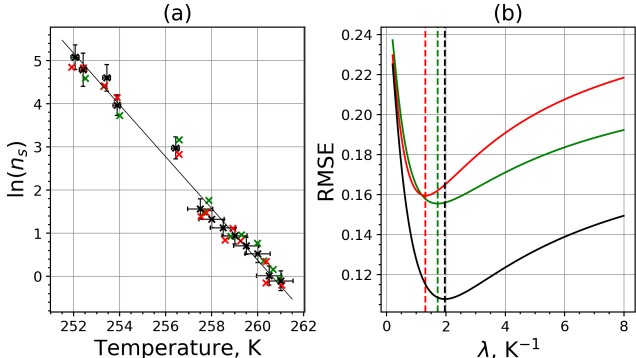

**Figure 11.** (a) Original (black), and two within the experimental error interval randomly generated data sets (red and green) using the measured INAS densities of kaolinite. The black solid line is the linear fit to the original experimental data set. (b) RMSE as the function of lambda for two of the 1000 randomly generated data sets. The vertical dashed lines indicate the optimal lambda values for the red, green and black curves.

was determined by fitting a linear regression curve to the log-linear graph, and subsequently calculating RMSE. We repeated this procedure 1000 times by generating new random data points, and from the statistical analysis of the obtained $\lambda$ values, $\Delta\lambda$

was calculated. As an example for the procedure, two randomly generated data sets (plotted in red and green colors) and the corresponding RMSE values as function of lambda are shown in Fig. 11b.

The optimal $\lambda$ value 1.7 was used to correct the temperature shift caused by the residence-time dependency of the freezing process in M-WT and by the cooling-rate dependency in M-AL. The corrected data points together with the fitted regression line are plotted in Fig. 9b. When comparing Fig. 9b to Fig. 9a, the agreement between the corrected data points from the two

distinct experimental methods is apparent. This is also supported by the high $R^2$ value of the regression line.

The temperature gradient of the normalized freezing rate, $\omega$, was already determined in Section 4.1 from the time dependency of the frozen fraction measured in M-WT. The data points corrected by the $\omega$-based temperature shift (i.e. if $\omega = \lambda$) are plotted in Fig. 9c together with the best fit line. Again, an obvious agreement can be seen between the two distinct experimental methods.

For a single-component INP, $\omega$ is equal to $\lambda$, which was found in Herbert et al. (2014) for their kaolinite sample from the Clay Mineral Society. In our study the $\omega = 0.49$ and $\lambda = 1.7$ values for our kaolinite sample from SigmaAldrich are differing. The deviation in the temperature correction based on $\lambda$ and $\omega$ is further emphasized in Fig. 9c where the regression lines obtained by employing the optimal $\lambda$ values and $\omega$ are plotted by dash-dotted and dashed lines, respectively. This plot suggests that the kaolinite sample investigated in our study is a multi-component system, and the determined $\lambda$-value should be employed for

correcting the measured freezing temperatures.

The $\omega$ and $\lambda$ values for the investigated materials are listed in Table 3. After definition of Herbert et al. (2014), all materials exhibit multiple-component behaviour since $\omega < \lambda$ in all cases. Nevertheless, for some materials, e.g., illite NX, despite dif-





**Table 3.** $\lambda$ and $\omega$ values, and the classification of the investigated materials. The results of statistical t-tests are also given: calculated t-values, number of samples (data points), and t-values showing significance in $\alpha = 99.5\%$.

| Material | $\lambda$ in K$^{-1}$ | $\omega$ in K$^{-1}$ | $t_{s,\omega}\,\vert$ | $t_{s,\lambda}\,\vert\,N\,\vert\,t_{sig}$ | Single/Multiple component |
|---|---|---|---|---|---|
| FC | $2.61 \pm 0.25$ | $1.41 \pm 0.33$ | $3.43\vert$ | $3.72\,\vert 26\vert\,3.725$ | Single |
| MCC | $1.57 \pm 0.04$ | $1.29 \pm 0.21$ | $-3.02\vert$ | $-2.03\,\vert 12\vert\,4.437$ | Single |
| Feldspar | $1.17 \pm 0.07$ | $0.65 \pm 0.09$ | $10.06\vert$ | $10.15\,\vert 39\vert 3.566$ | Multiple |
| Illite NX | $1.46 \pm 0.20$ | $0.87 \pm 0.16$ | $2.54\vert$ | $3.08\,\vert 28\vert\,3.689$ | Single |
| Kaolinite | $1.72 \pm 0.39$ | $0.49 \pm 0.03$ | $11.48\vert$ | $26.97\,\vert 13\vert\,4.318$ | Multiple |
| Montmorrilonite K10 | $1.43 \pm 0.21$ | $0.66 \pm 0.15$ | $5.46\vert$ | $7.03\,\vert 26\vert\,3.725$ | Multiple |
| Sahara dust SDB01 | $1.21 \pm 0.23$ | $0.84 \pm 0.09$ | $4.31\vert$ | $5.73\,\vert 16\vert\,4.073$ | Multiple |

ferent $\lambda$ and $\omega$ values the deviation between the data sets corrected using $\lambda$ or $\omega$ was not obvious (see Appendix A). To obtain further insights in this feature, we performed statistical significance tests as follows.

First, we computed the arithmetic mean curve of the two best fit lines and calculated their mean deviation $\bar{d}$ from that mean curve. The ultimate question of our statistics tests was, whether the mean deviation is significant with respect to the measurement error and data scatter. Hence, as the next step, the error weighted standard deviations of the residuals $s_\omega$ and $s_\lambda$ were calculated as

$$s_\omega = \sqrt{\frac{\sum_{i=1}^{N} \frac{\left(\Delta_{\omega,i} - \overline{\Delta_\omega}\right)^2}{\Delta T_i^2}}{\sum_{i=1}^{N} \frac{1}{\Delta T_i^2}}} \tag{19}$$

$$s_\lambda = \sqrt{\frac{\sum_{i=1}^{N} \frac{\left(\Delta_{\lambda,i} - \overline{\Delta_\lambda}\right)^2}{\Delta T_i^2}}{\sum_{i=1}^{N} \frac{1}{\Delta T_i^2}}} \tag{20}$$

where $\Delta T_i$ is the temperature measurement error, $\Delta_{\omega,i}$ and $\Delta_{\lambda,i}$ are the deviations of the data points from the corresponding best fit curves, while $\overline{\Delta_\omega}$ and $\overline{\Delta_\lambda}$ are the mean values of these deviations. For the significance test we applied a two-sided Student t-test on a significance level of 99.9%, and calculated

$$t_{s,\omega} = \frac{|\bar{d} - \mu_0|}{s_\omega} \cdot \sqrt{N} \tag{21}$$

$$t_{s,\lambda} = \frac{|\bar{d} - \mu_0|}{s_\lambda} \cdot \sqrt{N} \tag{22}$$

where $N$ is the number of data points. The null hypothesis was that the two linear curves do not significantly differ, thus, $\mu_0 = 0$ for their deviation from the arithmetic mean curve. In Table 3 we listed the calculated $t_{s,\omega}$ and $t_{s,\lambda}$, the number of





data points, and the tabulated $t_{sig}$ ($\beta = 99.9\%$) values for the Student t-test for each material. If $t_{s,\omega}$ or $t_{s,\lambda}$ is greater than $t_{sig}$ ($\beta = 99.9\%$), then the null hypothesis is rejected. That means that the two best fit lines are differing significantly with respect

to data scatter and measurement error, and consequently, the material is treated as multiple component on a 99.9% confidence level. Otherwise we consider the material as single component albeit the statistical test does not prove the null hypothesis. Hence, we classify the material as single or multiple component within our measurement error and data scatter.

As listed in Table 3, according to our statistical test kaolinite, feldspar, montmorillonite, and Sahara dust are multiple-component, while illite NX, FC, and MCC are single-component INP. This implies that the definition of Herbert et al. (2014)

to distinguish between single and multiple component samples on the basis of $\lambda$ and $\omega$ values cannot directly be applied to our M-AL and M-WT experiments. This is the consequence of the adaptive cooling of the drops in M-AL which results in a temperature dependence in the $\lambda$ based correction. Thus, the same $\lambda$ value caused a higher temperature correction at higher temperatures (see Fig. B4 in the Appendix). Therefore, our analysis indicates that statistic tests have to be performed considering both data scatter and measurement error to compare the $\lambda$ and $\omega$ values. This procedure improves the classification

of the materials as single or multiple component.

The statistical tests supported that the kaolinite which we analysed is multiple component. That contradicts the finding of Herbert et al. (2014) who showed their kaolinite sample (KGa-1b from the Clay Mineral Society) to be single component with $\lambda = \omega = 1.12$. This indicates that these two kaolinite samples are different and, thus, the result outputs cannot directly be compared since the IN activity of materials depends on their specific chemical composition, which is known to be very variable

for kaolinite. For example, the $\lambda$ value for the kaolinite used in the cooling experiments of Wright and Petters (2013) was 1.7, which is equal to our result. In contrast, the Fluka kaolinite sample measured by Welti et al. (2012), which is known to contain particles of very ice active feldspar, had a $\lambda$ value of 2.2 (see Table 2 in Herbert et al. (2014)). In general, we found slightly higher $\lambda$ values for biological aerosols (FC and MCC) than for mineral dusts. This results in smaller $\Delta T$ in Eq. (12), and hence, biological INP show a weaker cooling rate dependence, in agreement with the findings of Peckhaus et al. (2016). The

temperature correction ranged for the investigated samples in our experiments from $\approx 0.5\,\mathrm{K}$ up to several Kelvins, depending on the material's $\lambda$-value (see also Fig. B4 in the Appendix).

The composite plot of the INAS densities for all investigated materials obtained by M-WT and M-AL measurements is shown in Fig. 12. In accord with the literature (e.g., (Atkinson et al., 2013)), feldspar is far the most efficient ice nucleating particle type among the investigated materials. Besides feldspar, kaolinite has also a high IN efficiency, in particular at higher

temperatures. The biological particles (FC, MCC) and the clay minerals illite NX and Sahara dust have similar temperature dependent $n_s$ values. The one exception is montmorillonite, which was found to be the least efficient within the investigated temperature range from $248\,\mathrm{K}$ to $266\,\mathrm{K}$. In Fig. 12 also shown are parameterizations for feldspar ((Atkinson et al., 2013)) and for desert dust ((Ullrich et al., 2017)). Our temperature corrected feldspar data fits very well to the parameterization of Atkinson et al. (2013) which was based on cold stage experiments, i.e. using aqueous suspensions of INP material. In contrast, the desert

dust parameterization of Ullrich et al. (2017) is based on dry deposition experiments and predicts higher INAS densities as measured in M-WT and M-AL. This is in accord with the literature, as for example Hiranuma et al. (2019) revealed different INAS densities when dry deposition or aqueous suspension techniques were utilized.





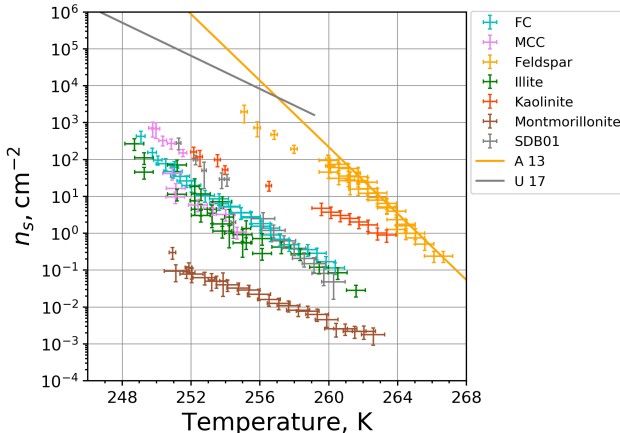

**Figure 12.** INAS densities of the investigated materials as function of temperature. The data points are composites from M-WT and M-AL measurements and are corrected for the cooling rate. Orange and gray solid lines show parameterizations for feldspar ((Atkinson et al., 2013)) and desert dust ((Ullrich et al., 2017)), respectively.

## 6  Conclusions and Suggestions

Immersion freezing efficiencies of different types of aerosol particles, such as pure and natural clay minerals as well as biolog-
ical particles, were studied using two distinct measurement techniques, an acoustic levitator (M-AL) and a vertical wind tunnel
(M-WT). Both instruments utilize freely floating of individual droplets.

The INAS densities of different types of aerosol particles obtained by M-AL and M-WT revealed a shift in the freezing tem-
peratures to lower values. Such a shift in freezing temperatures became obvious in our earlier experiments in the measurement
campaigns FIN02 (DeMott et al., 2018) and INUIT-cellulose (Hiranuma et al., 2019). Therefore, we had already corrected the
data published in those papers for the freezing temperature shift. Following the procedure depicted in Fig. 10, we were able to
bring the INAS densities obtained from the two different methods in line. We have also reconciled our earlier experiments on
illite NX (Diehl et al., 2014) and ascertained that those data were burdened with a temperature shift as well. A modification of
the data in Diehl et al. (2014) according to our new findings improves further the agreement of the data from M-WT and M-AL
(green symbols in Fig. 12).

Taking advantage of having two independent single droplet levitation methods located in our laboratory we determined the
material dependent temperature correction factor for the investigated aerosol types based on the analysis method suggested
by Herbert et al. (2014). Furthermore, we classified the aerosol materials investigated in this study as single- or multiple-
component.

An important conclusion on the applicability of laboratory-based immersion freezing measurement techniques can be made
due to the different air flow conditions applied in our experiments. In M-WT a continuous air flow is established around a
floating droplet (correctly simulating the real atmospheric condition), whereas M-AL maintains levitation with a very weak



airflow. Since the INAS densities obtained by M-WT and M-AL after temperature correction show very good agreement, one can conclude that the airflow around the droplets containing the INP does not significantly influence the immersion freezing process.

Based on the experiences collected during the presented synergetic study we suggest the following points for future immersion freezing studies:

– If the instrument used for the measurements utilizes a continuously varying cooling rate, then its temperature adaptation decay has first to be characterized in terms of equilibrium temperature and decay constant and the corresponding uncertainties. Furthermore, the drop temperature has to be measured directly, because it can significantly deviate from the ambient temperature.

– When comparing the IN efficiencies measured by different instruments utilizing distinct cooling rates, the comparison has to be carried out very carefully and critically. We suggest to use the same cooling rate in the different instruments in such intercomparison studies.

– We note from Fig. 8 that freezing behavior and, consequently, the necessity of background correction depends on the purity grade and age of water used for producing aqueous suspension samples. Therefore, it has to be carefully characterized for all experiments, as well.

– In case the INAS densities are measured applying a non-standard cooling rate (i.e. $CR \neq 1\,\mathrm{K\,min^{-1}}$), the freezing temperatures have to be corrected following the procedures of Herbert et al. (2014) and the one described in this study. It has to be taken into account that the temperature correction is material-dependent, and most probably temperature dependent for most of the INP.

– By the characterization of the aerosol material in terms of temperature correction, statistical significance tests should be carried out taking both the data scatter and the measurement error into account. Of course, by increasing the measurement sensitivity (i.e. decreasing the measurement error) or by decreasing the data scatter (either by improving measurement accuracy, or due to less natural variability of the sample material), the prediction whether the material is single or multiple component will be more accurate. Nevertheless, the classification can only be obtained within the measurement error and accuracy of the applied experimental method.

– In cloud models the cooling rate has to be considered and the freezing temperatures of materials have to be corrected taking the material dependent $\lambda$-values into account.

– The $\lambda$-based temperature shifts of some aerosol species determined by Herbert et al. (2014) and in the present study serve rather for orientation. We suggest the determination of such temperature shifts for the specific material samples under investigation in each future experimental study on immersion freezing of aerosol particles.

*Data availability.* The measurement data will be provided upon request.



*Video supplement.* A video supplement showing the record of the immersion freezing of a liquid drop in the M-AL can be downloaded from

https://doi.org/10.5446/46729




## Appendix A: Further reconciled experimental results

In this section we provide the experimental results for the determination of the temperature dependent freezing rate (as in Fig. 6), as well as the composite INAS density spectra from M-WT and M-AL using $\lambda$ and $\omega$ (as in Fig. 9) for the materials listed in Table 1.

### A1  Fibrous cellulose (FC)

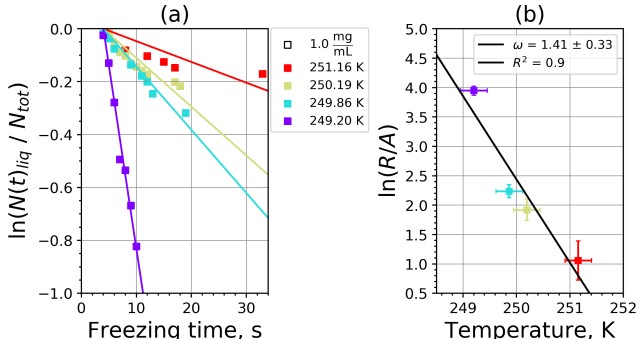

**Figure A1.** (a) The decrease of fraction of droplets which remained liquid with time at different temperatures in the isothermal experiments of fibrous cellulose (FC) at the M-WT. The colors are corresponding to different temperatures (particle concentrations $1\,\mathrm{g\,L^{-1}}$). Typical error bars are depicted in Fig. 6. (b) Freezing rate of FC normalized to surface area as function of temperature.

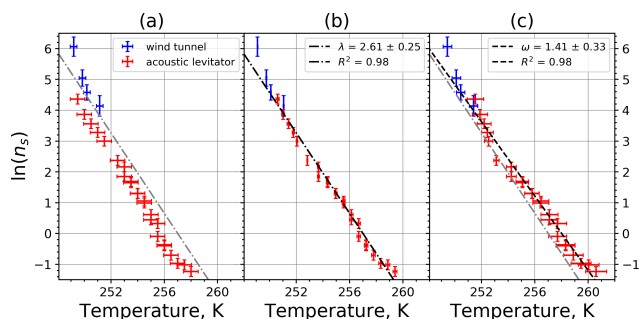

**Figure A2.** (a) Composite INAS density spectrum of FC from the uncorrected M-WT (blue) and M-AL (red) measurements. (b) and (c) show the temperature corrected data points from the M-WT and M-AL experiments based on $\lambda$ and $\omega$, respectively. The dash-dotted line in (c) is the regression line on corrected data points obtained by employing the optimal $\lambda$-value as in (b).





## 550  A2   Microcrystalline cellulose (MMC)

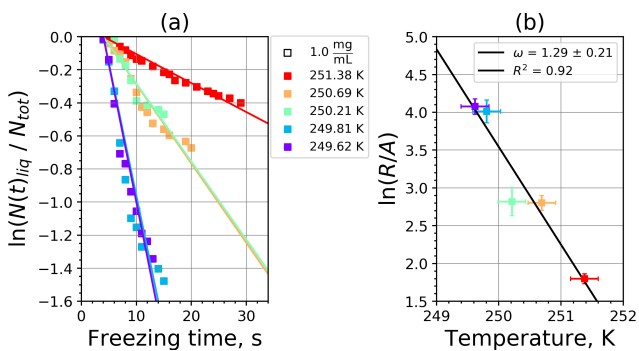

**Figure A3.** (a) The decrease of fraction of droplets which remained liquid with time at different temperatures in the isothermal experiments of microcrystalline cellulose (MCC) at the M-WT. The colors are corresponding to different temperatures (particle concentrations $1 \, \mathrm{g \, L^{-1}}$). Typical error bars are depicted in Fig. 6. (b) Freezing rate of MCC normalized to surface area as function of temperature.

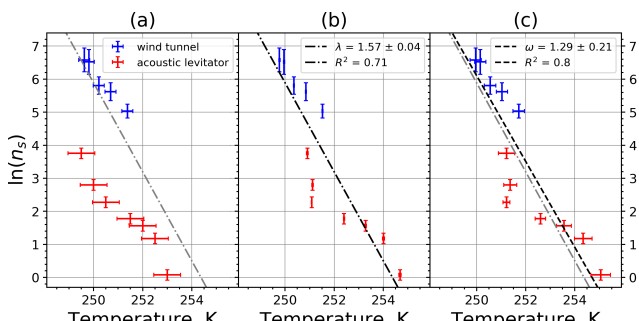

**Figure A4.** (a) Composite INAS density spectrum of MCC from the uncorrected M-WT (blue) and M-AL (red) measurements. (b) and (c) show the temperature corrected data points from the M-WT and M-AL experiments based on $\lambda$ and $\omega$, respectively. The dash-dotted line in (c) is the regression line on corrected data points obtained by employing the optimal $\lambda$-value as in (b).





## A3 Feldspar

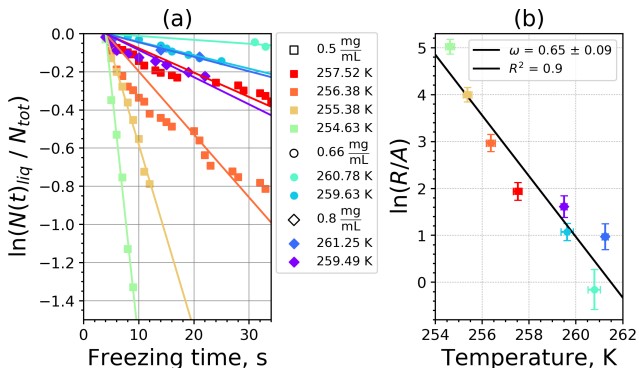

**Figure A5.** (a) The decrease of fraction of droplets which remained liquid with time at different temperatures in the isothermal experiments for feldspar at the M-WT. The colors are corresponding to different temperatures; experiments with particle concentrations of $0.5\,\mathrm{g\,L^{-1}}$, $0.66\,\mathrm{g\,L^{-1}}$, and $0.8\,\mathrm{g\,L^{-1}}$ are plotted by rectangles, circles and triangles, respectively. Typical error bars are depicted in Fig. 6.(b) Freezing rate of feldspar normalized to surface area as function of temperature.

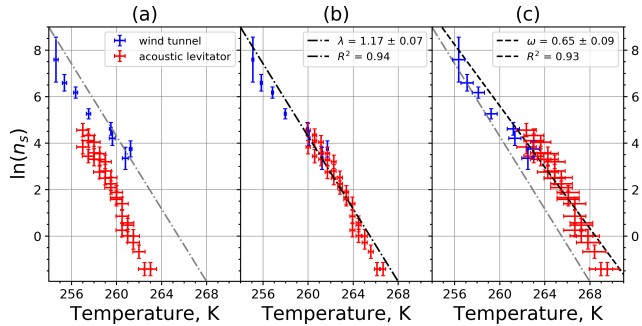

**Figure A6.** (a) Composite INAS density spectrum of feldspar from the uncorrected M-WT (blue) and M-AL (red) measurements. (b) and (c) show the temperature corrected data points from the M-WT and M-AL experiments based on $\lambda$ and $\omega$, respectively. The dash-dotted line in (c) is the regression line on corrected data points obtained by employing the optimal $\lambda$-value as in (b).



## A4  Illite NX

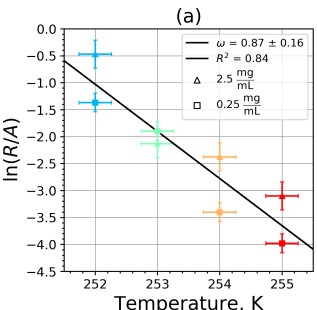

**Figure A7.** Freezing rate of illite NX normalized to surface area as function of temperature for experiments at the M-WT with particle concentrations of $2.5\,\mathrm{g\,L^{-1}}$ and $0.25\,\mathrm{g\,L^{-1}}$ (plotted by triangles and rectangles, respectively). Freezing rates were calculated from the time dependence of the liquid ratio of illite NX presented in Fig. 6. in Diehl et al. (2014).

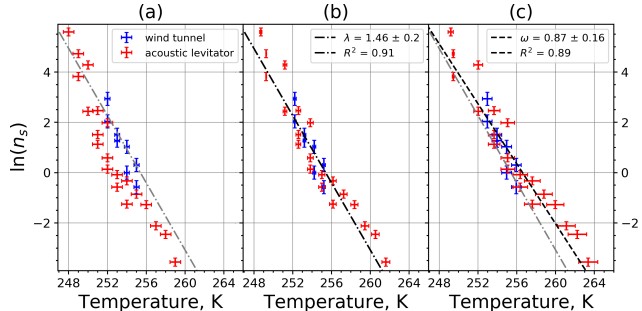

**Figure A8.** (a) Composite INAS density spectrum of illite NX from the uncorrected M-WT (blue) and M-AL (red) measurements. (b) and (c) show the temperature corrected data points from the M-WT and M-AL experiments based on $\lambda$ and $\omega$, respectively. The dash-dotted line in (c) is the regression line on corrected data points obtained by employing the optimal $\lambda$-value as in (b).




## A5 Montmorilonite

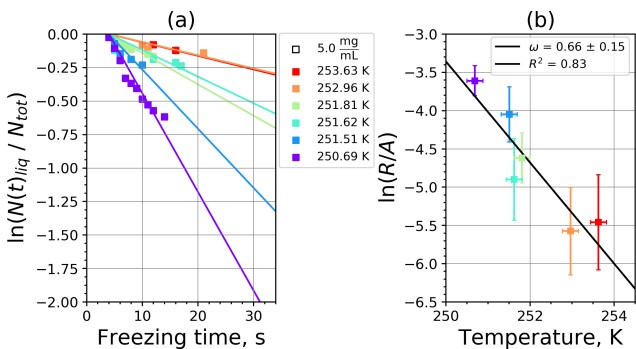

**Figure A9.** (a) The decrease of fraction of droplets which remained liquid with time at different temperatures in the isothermal experiments for montmorillonite at the M-WT. The colors are corresponding to different temperatures (particle concentrations $5\,\mathrm{g\,L^{-1}}$). Typical error bars are depicted in Fig. 6.(b) Freezing rate of montmorillonite normalized to surface area as function of temperature.

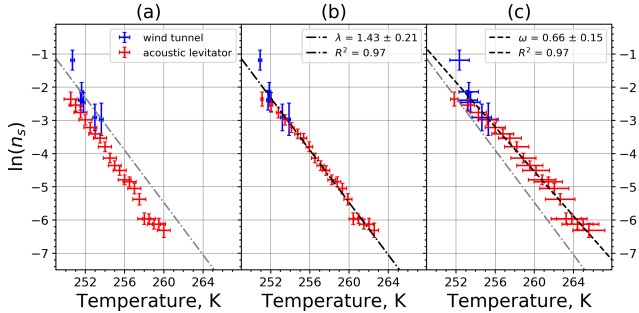

**Figure A10.** (a) Composite INAS density spectrum of montmorillonite from the uncorrected M-WT (blue) and M-AL (red) measurements. (b) and (c) show the temperature corrected data points from the M-WT and M-AL experiments based on $\lambda$ and $\omega$, respectively. The dash-dotted line in (c) is the regression line on corrected data points obtained by employing the optimal $\lambda$-value as in (b).



## Appendix B: Drop temperature adaptation in the M-AL and in the M-WT

A liquid droplet placed in a colder or warmer environment tends to a quasi-steady state temperature difference between itself and its surrounding. In order to describe the temperature adaptation process, diffusional and convective heat and mass transfers for water vapor are considered. We will follow the concept of Pruppacher and Klett (2010) in the forthcoming derivation. Hence, first, the heat and mass transfer of a motionless droplet will be described, and after that the effect of air ventilation will be introduced. The symbols used here are listed in the List of Symbols.

### B1   Diffusional heat and mass transfer of a motionless drop in equilibrium

When computing the simple case of the diffusional heat transfer of a motionless water droplet in air latent heat from condensation or evaporation is not considered. The rate of heat is calculated by integrating the heat flux density over the entire droplet surface. The heat flux density can be derived from Fourier's law which in spherical coordinates reads as,

$$j_{h,r}|_{r=a} = -k_a \left( \frac{\partial T}{\partial r} \right)_{r=a} \tag{B1}$$

where $r$ is the distance from the drop center. Thus, the rate of heat transfer of a motionless drop considering pure diffusional heat transfer is

$$\left( \frac{dq}{dt} \right)_0 = -k_a \int_S \left( \frac{\partial T}{\partial r} \right)_{r=a} dS \tag{B2}$$

The temperature is determined by solving the heat conduction equation which has its form in spherical coordinates for a motionless drop under steady state thermal condition:

$$\frac{\partial^2 T}{\partial r^2} + \frac{2}{r} \frac{\partial T}{\partial r} = 0 \tag{B3}$$

This partial differential equation is solved using the boundary conditions,

$$T(r = \infty) \quad = \quad T_\infty \tag{B4}$$

$$T(r = a) \quad = \quad T_a \tag{B5}$$

where $T_\infty$ is the temperature in the free air, i.e. far away from the drop; and $T_a$ is the drop surface temperature, while $a$ is 575 the drop radius. The solution for the temperature as function of $r$ is,

$$T(r) = T_\infty + (T_a - T_\infty) \frac{a}{r} \tag{B6}$$

Hence,

$$\left( \frac{dq}{dt} \right)_0 = 4\pi a k_a (T_\infty - T_a) \tag{B7}$$

Similarly to the heat transfer, the mass transfer rate of an motionless droplet in equilibrium with its surrounding air is calculated 580 from

$$\left( \frac{dm}{dt} \right)_0 = -D_v \int_S \left( \frac{\partial \rho_v}{\partial r} \right)_{r=a} dS \tag{B8}$$





where $D_v$ is the water vapor diffusion coefficient, and $\rho_v$ is the water vapor density in the surrounding air around the water droplet. The water vapor density can be found by solving the convective diffusion equation,

$$\frac{\partial \rho_v}{\partial t} + \nabla \rho_v u = D_v \nabla^2 \rho_v \tag{B9}$$

This differential equation simplifies for a motionless drop in steady state to Laplace's equation in the form: $\nabla^2 \rho_v = 0$ (cf. B3).The boundary conditions for the problem are

$$\rho_v(r = \infty) = \rho_{v,\infty} \tag{B10}$$

$$\rho_v(r = a) = \rho_{v,a} \tag{B11}$$

where $\rho_{v,\infty}$ and $\rho_{v,a}$ are the water vapor densities in the air far away from the drop and at the drop surface, respectively. The
solution of the governing differential equation is similar to that for the heat transfer. Thus, the rate of change of the mass of a motionless droplet due to diffusion of water vapor under steady state conditions is given by,

$$\left(\frac{dm}{dt}\right)_0 = 4\pi D_v \left(\rho_{v,\infty} - \rho_{v,a}\right) \tag{B12}$$

**B2  Heat and mass transfer of an evaporating drop in air flow**

We now consider the more realistic and atmospherically relevant case involving also the effect of air motion around the droplet
The total rate at which a drop falling in air gains heat is the sum of the convective heat flux from the air to the drop and the heat loss of the drop by releasing latent heat due to evaporation:

$$\left(\frac{dq}{dt}\right)_a = 4\pi a k_a \left(T_\infty - T(t)\right) \cdot f_h + L_e \frac{dm}{dt} \cdot f_v \tag{B13}$$

where the so called ventilation coefficients $f_h$ and $f_v$ are introduced accounting for the enhanced heat and mass transfer, respectively, due to ventilation. Thus, for a drop in an air flow $f_h > 1$ and $f_v > 1$, while a motionless evaporating drop can be
described using Eq. (B13) by setting $f_h = 1$ and $f_v = 1$. In M-AL or in M-WT where a warm ($\approx 20\,^\circ\text{C}$) drop is injected into a cold subsaturated environment, both terms on the right hand side of Eq. (B13) are negative. The drop cools down at a rate proportional to its mass $m$ determined by

$$\left(\frac{dq}{dt}\right)_a = m c_w \frac{d}{dt} \left(T_a - T_\infty\right) \tag{B14}$$

where $c_w$ is the specific heat capacity of water. By equating Eqs. (B13) and (B14) we get the governing equation for the
temperature adaptation of the droplet as

$$\frac{4\pi}{3} a^3 \rho_w c_w \frac{d}{dt} \left(T_\infty - T_a(t)\right) = -4\pi a k_a \left(T_\infty - T_a(t)\right) \cdot f_h$$
$$-4\pi a L_e D_v \left(\rho_{v,\infty} - \rho_{v,a}\right) \cdot f_v \tag{B15}$$





After integration, we obtain the following solution of this differential equation:

$$T_\infty - T_a(t) - \delta = (T_\infty - T_a(t=0) - \delta)\exp(-t/\tau) \tag{B16}$$

with the time constant

$$\tau = \frac{a^2 \rho_w c_w}{3\left[k_a L_e D_v \left(\frac{d\rho_v}{dT}\right)_{sat}\right]\cdot f_h} \tag{B17}$$

and

$$\delta = \frac{D_v L_e f_v \left[\frac{1-r_v}{r_v}\rho_{v,\infty}\right]}{k_a \cdot f_h + L_e D_v f_v \left(\frac{d\rho_v}{dT}\right)_{sat}} \tag{B18}$$

which gives the steady temperature difference between the equilibrium temperature ($T_e = T_a(t \to \infty)$) of a ventilated evapo-

rating drop and its surrounding air at a relative humidity of $r_v$. For simplicity we did not indicate but the physical quantities
are represented by their averages over the integration interval. Furthermore, we assumed that $f_h = f_v$ (Pruppacher and Klett,
2014).

After some manipulation, and using $\rho_{v,sat}(T_e) = \rho_{v,a}(T_e)$ one can get two other forms for $\delta$:

$$\delta = \frac{D_v L_e f_v [\rho_{v,sat}(T_e) - \rho_{v,\infty}]}{k_a \cdot f_h} \tag{B19}$$

or, by applying the ideal gas law

$$\delta = \frac{D_v L_e f_v}{k_a \cdot f_h} \frac{M_w}{R}\left(\frac{e_{sat}(T_e)}{T_e} - \frac{e_\infty}{T_\infty}\right) \tag{B20}$$

where $e_{sat}(T_e)$ is the saturation water vapor pressure at temperature $T_e$, and $e_\infty$ is the water vapor pressure in the air. Hence,
the equilibrium drop temperature is given as,

$$T_e = T_\infty - \frac{D_v L_e f_v}{k_a \cdot f_h} \frac{M_w}{R}\left(\frac{e_a(T_e)}{T_e} - \frac{e_\infty}{T_\infty}\right) \tag{B21}$$

In M-AL the levitating drop may gain heat from the absorbed acoustic energy at a certain constant rate:

$$\frac{dq_{ac}}{dt} = e_{ac}V \tag{B22}$$

where $e_{ac}$ is the acoustic energy density flux and $V$ is the drop volume. $e_{ac}$ varies with time at a very high frequency ($\approx 20$
kHz) therefore it can be considered as time independent when discussing the slow process of heat transfer. Since this term
is independent on temperature and time, it does not affect $\tau$ and will not appear in Eq. (B17), which describes the time

dependence of the temperature adaptation process. Nevertheless, the absorbed acoustic energy heats up the drop and increases
$\delta$ with a constant temperature value. This temperature difference between the theoretically calculated equilibrium temperature
and the environmental temperature was also observed in M-AL, and calculated to be $\approx 4.5K$.



## B3 Drop surface temperature in M-AL

Although there is seemingly no air flow around a drop levitating in M-AL, the pressure distribution caused by the acoustic
waves does create convection about it (Lierke, 1995). This has to be considered in the temporal evolution calculation in
Eq. (B17). In Fig. B1 an example of the measured surface temperature evolution of a 2-mm diameter drop placed into M-AL
is plotted by black line. Neglecting ventilation around the drop (i.e. $f_v = 1.0$ in Eq. (B17)) the cooling would be much slower
(blue line) than in reality. Setting the ventilation coefficient to $f_v = 5.2$ – which value is close to $f_v = 3$ determined by Lierke
(1995) – the temperature evolution follows accurately the measured curve (green line).

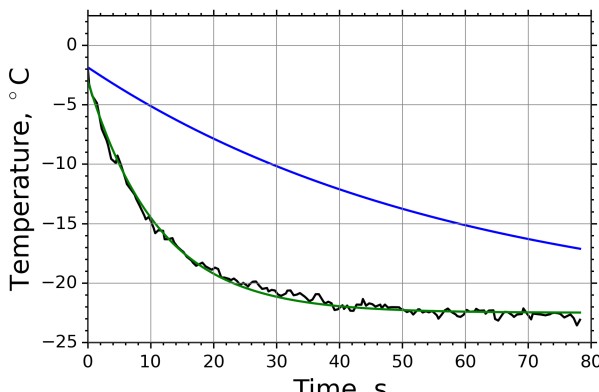

**Figure B1.** Temporal evolution of the surface temperature of a 2-mm diameter drop injected into M-AL: measured (black line); calculated
using Eq. (B16) without ventilation (blue line); calculated using Eq. (B16) with a ventilation factor $f_v = 5.2$ (green line).

## B4 Internal drop temperature simulation in M-AL

In M-AL experiments the continuous sharp surface temperature drop caused by the adaptation to the significantly colder
environment results in a temperature difference in the drop interior. Therefore, the temperature at the drop surface is lower
than close to the drop center. Since the drop temperature was determined in the M-AL experiments by measuring the surface
temperature by means of an infrared thermometer, the actual (internal) temperature experienced by ice nucleating particles
inside the drop is higher than the measured value. This measurement artifact might falsify the experimentally determined
temperature dependence of the ice nucleating ability. In order to estimate this experimental issue, a simulation was carried
out based on the theoretical formulation of the temperature adaptation given above and on heat conduction inside the liquid
drop. The drop volume was split into 10 layers of equivalent radii and the heat conduction among the layers was calculated by
solving the transient heat equation

$$\frac{\partial T}{\partial t} = \frac{\partial^2 T}{\partial r^2} + \frac{2}{r}\frac{\partial T}{\partial r} \qquad (B23)$$





For the numerical integration explicit finite difference method was used. During the simulation the surface temperature was continuously cooling following the experimentally obtained adaptation curve shown in Fig. B1. The temperature distribution inside the drop is depicted at four time instances (i.e. drop surface temperatures) in Fig. B2 revealing the temperature difference between the drop surface and the drop center.

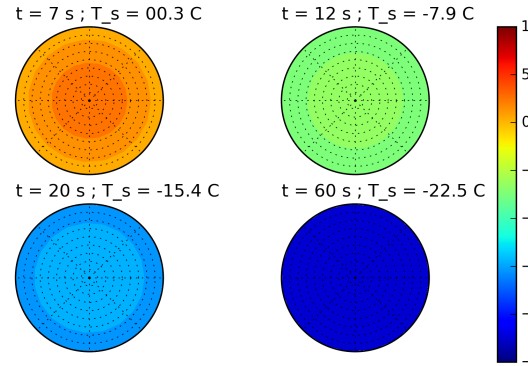

**Figure B2.** Drop internal temperatures at four different time instances for a continuously cooling drop in M-AL following Fig. B1, based on numerical simulation of the heat conduction equation.

The temperature variation of three layers together with the volume averaged drop temperature change relative to the surface temperature are further investigated in Fig. B3. While the temperature at drop's center (red curve) deviates from the surface temperature by up to $2.5\,°C$, the second outmost layer representing 40% of the entire drop volume (green curve) follows the surface temperature within $0.5\,°C$ at subzero temperatures. At temperatures below $-5\,°C$, which are relevant for immersion freezing experiments, the temperature difference is less than $0.3\,°C$, thus, within the measurement uncertainties. The volume averaged drop temperature (magenta curve in Fig. B3) is also within the measurement uncertainty of $0.5\,°C$ for this temperature range.

The simulation was carried out without considering any internal circulation, which would further and faster unify the temperature distribution inside the liquid. Considering the large number of ice nucleating particles ($\gtrsim 300.000$) immersed in each of the drops, one can conclude, that the surface temperature measured by the pyrometer can be used as characteristic drop temperature.

The temperature difference between the surface and the volume average temperature of the drop is compared to the temperature shift calculated for three different $\lambda$ values in Fig. B4. The figure reveals that the calculated temperature difference inside the drop (magenta curve) is a factor of 6 to 20 smaller than the temperature shift caused by the high cooling rate in M-AL for different $\lambda$ values (red, green, and blue lines). Therefore, this effect cannot explain the observed freezing temperature shift in M-AL.





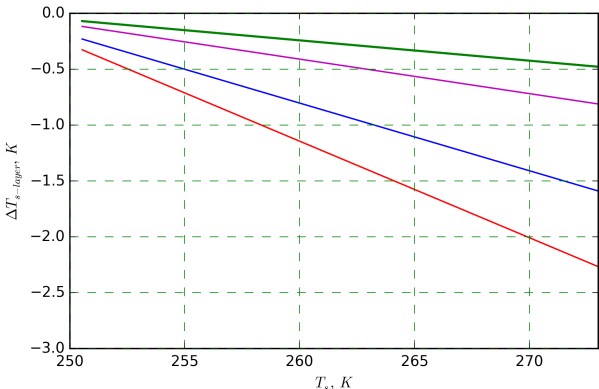

**Figure B3.** Temperature difference between the drop surface and internal drop layers calculated from the numerical simulation: drop center (red curve); layers number 5 (blue) and 8 (green) representing 20% and 60% of entire drop volume, respectively; and volume average temperature (magenta).

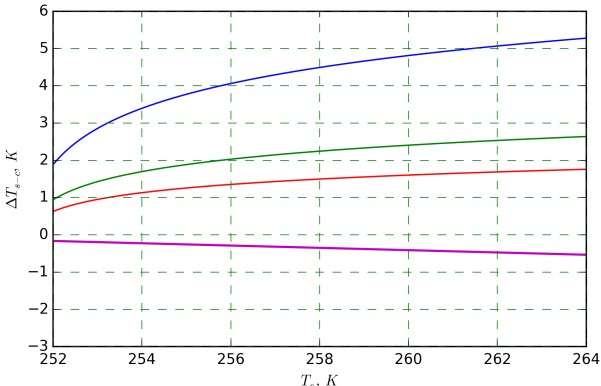

**Figure B4.** Temperature shift calculated for three different $\lambda$ values (red: 0.8, green: 1.6., and blue: 2.4) in relation to temperature difference between drop surface and volume averaged drop temperature (magenta curve).

*Author contributions.* MS wrote the paper with assistance from all co-authors contributing comments on results, discussion and conclusion; MS analysed the data, performed the numerical simulations, and carried out M-WT experiments on feldspar, FC, and MCC; MS and SKM conceived the M-AL and M-WT experiments; M. D. performed M-AL experiments on illite NX, kaolinite, and montmorillonite and evaluated the results; A. M. performed M-AL experiments on FC, and MCC, and evaluated the results; C.P.L. performed M-AL and M-WT experiments on feldspar and SDB01; O.E. performed M-AL and M-WT experiments on FC, MCC, illite NX, and feldspar.




*Competing interests.* The authors declare that they have no conflict of interest.

*Acknowledgements.* The authors acknowledge support from Deutsche Forschungsgemeinschaft (DFG) under contract SZ260/4-2 within Research Unit INUIT (FOR 1525). We also thank Prof. Holger Tost for his help in the statistical test methodology.

**List of Symbols**

The next list describes several symbols that will be later used within the body of the document

$\beta$        freezing temperature shift for water vapor (Vali, 2014)

$\delta N_{fr}$   number of droplets freezing within time interval $\delta t$

$\Delta T_{iso}$   temperature shift in isothermal experiments due to a relative change in residence time

$\Delta T_r, \Delta T_f$ absolute and normalized temperature shifts, respectively, in cooling experiments due to any change in cooling rate

$\delta$        Temperature difference between a drop and its environment in equilibrium (i.e., $\delta = T_\infty - T_e$))

$\lambda$        Temperature gradient of the heterogeneous nucleation rate coefficient, $J_s$

$\omega$        Temperature gradient of the freezing rate

$\tau$        time constant of the temperature adaptation of a drop placed in cold/warm environment

$c$        particle mass concentration in the sample solution

$f_ice$   cumulative fraction of droplets frozen between 0 °C and temperature $T$

$J_s, J_{s,i}$ heterogeneous nucleation rate coefficient of a single component system, and of subpopulation $i$

$J_{hom}$   homogeneous nucleation rate coefficient

$n_s$        ice nucleation active site density

$n_{fr}, n_{liq}$ number of frozen and liquid droplets in a freezing experiment, respectively

$N_{tot}$        total number of droplets in the population

$R(T)$   Freezing rate at a fixed temperature

$R(t,T)$  Rate of supercooled droplets freezing per unit time at a fixed temperature

$r$        Cooling rate in the experiments

$S_p$      total particle surface area

$s_\omega\, s_\lambda$      error weighted standard deviations of the residuals for $\omega$ and $\lambda$

$SSA$      specific surface area of the particle

$T$      Temperature

$t$      measurement time

$T_a$      Drop surface temperature

$T_e$      Equilibrium temperature between ventilated evaporating droplet and its environment

$T_\infty$      Air/environmental temperature

$t_{s,\omega}\, t_{s,\lambda}$      t-numbers corresponding to $\omega$, and $\lambda$ calculated for applying a two-sided Student t-test

$t_{sig}$      t-number corresponding to a significance level of $99.9\%$ in a two-sided Student t-test

$V_d$      aqueous suspension drop volume



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
