# Peer review of "Comparative Study On Immersion Freezing Utilizing Single Droplet Levitation Methods"

_Atmospheric Chemistry and Physics, 2020_

## Referee Comment (RC1) · Anonymous Referee #1 · 31 Aug 2020

**1  Overview**

The manuscript submitted to *Atmospheric Chemistry and Physics* titled "Comparative Study On Immersion Freezing Utilizing Single Droplet Levitation Methods" by Szakáll et al. presents an ice nucleation study of a variety of different types of particles using two methods, a vertical wind tunnel (WT) and acoustic levitator (AT). The WT gives approximate isothermal conditions and the AT has a cooling rate. The authors claimed if a particle type was single or multiple component by comparing values of ice active surface site densities, $n_s$, of these isothermal and cooling rate experiments, and following the derivation of Herbert *et al.* (2014). They conclude that freezing temperatures should be shifted with respect to derived $n_s$ values. In addition, there is a list of sug-
gestions for further study. This study has value to the *Atmospheric Chemistry and Physics* community providing new $n_s$ values for a variety of particle types, and parameters to describe their experiments following a single or multiple component approach. The methodology combines careful and well designed cooling rate and isothermal experiments, which is certainly relevant for evaluating time and temperature dependence for ice nucleation. There is clear uncertainty and statistical analysis for their conclusions. Unfortunately, there are major issues that must be resolved before I can recommend publication. First, there is insufficient review of previous studies that makes the manuscript unbalanced. Additionally, assumptions about surface area and the impact of surface area calculations on their conclusions are not stated. There is a fundamental flaw with shifting observed temperatures outside their investigated range. Finally, it must be acknowledged, that their conclusions about a temperature shift or a particle type being single or multiple component is dependent entirely on their choice of data analysis procedure and not a fundamental property of the types of particles.

**2 Major Issues**

- Descriptions of specific previous studies and their main claims using isothermal and cooling rate conditions are not included, but must be in order to be fair and balanced. Herbert *et al.* (2014) is extensively referenced for its data analysis, however it must be mentioned that they performed isothermal and cooling rate experiments on same particle types. Older studies such as Vali (1994) have cooling rate and isothermal conditions in a single experiment and support a multiple component approach held by the authors. Other studies, such as Alpert and Knopf (2016) and Knopf *et al.* (2020) come to a very different conclusions than the authors of the manuscript, but also used isothermal and cooling rate experiments. In summary, the literature review is incomplete and must be modified to include these references and any other relevant studies the authors wish to

include.

- On l. 113, the authors state that in the case of interparticle variability, Eqn 3 cannot be used and $J_s$ is modified (Eqn 9) as a result. Eqn 3 also has surface area, $S_p$, which has its own uncertainty and variability. What is the estimated error on $S_p$ and how does this error impact their findings? It is important the authors claim that the surface area in each droplet was not directly measured (only calculated from Eqn 4), and so the authors cannot rule out that droplet to droplet variability may be more than they expect.

  Along the same lines, the authors choose to vary cooling rate, however, varying surface area could also be done to test for single or multiple components. For example, Hartmann *et al.* (2016) changed particle mobility diameter from 0.3-0.7 $\mu$m and found similar $n_s$ values for kaolinite. It would follow that kaolinite would be considered as a single component particle type under sizes relevant for the atmosphere, in contrast to this study in which the authors labeled kaolinite as multiple component. In light of the work of Hartmann *et al.* (2016), I would ask the authors to include some discussion about changing particle size and the impact on what could be declared as multiple or single component. The authors should state the upper size limit of single particles in their droplets for each type.

- Another major concern is that the formulation here and in Herbert *et al.* (2014) is purely empirical and not based on any physical theory. Therefore, a particle type declared by the authors as a single and multiple component, or having a material dependent temperature correction factor (shift) is also empirical and not a fundamental property. This is a necessary caveat that must be stated. I do not wish to discount the evidence the authors give for either, however, they do need to stress that there is no direct observation of single or multiple components.

- Applying a temperature shift is fundamentally flawed. For example, the warmest observed temperature that feldspar nucleates in this manuscript is about 263 K.

However, the temperature correction is made to 269, which is 5 K warmer than measured. The authors did not measure nucleation at this warmer temperature. This is extrapolating data outside of the observed temperature range. Why don't the authors shift $n_s$ instead? Afterall, temperature is measured. $n_s$ is not measured, but it is calculated from measurements. This is a caveat that must be claimed and the reader must be warned about using data outside of the authors measured temperature range.

**3  Minor Issues**

- There are frequent typos and instances of comma misuse. In addition, there are multiple instances of greek letters being spelled out instead of actually using the symbol. This lack of proofreading is not appropriate for a manuscript submission and shows a lack of care for their own work. I urge the authors to correct this.

- l. 11-12. In the abstract, the authors write the words "material dependent correction factor". However, it is not claimed what is incorrect. That would be helpful in the abstract. The measured temperature is not incorrectly measured, so please be precise and tell the reader what is really being corrected.

- l. 20. Please change "The nucleation abilities..." to "The ice nucleation abilities..."

- l. 25-27. This is an inaccurate sentence. $n_s$ is calculated from the total number of nucleation events per unit surface area of the particles, and then it is assumed to be equivalent to the number of sites on the particles. This is a big difference to what is written. The point here is to stress the fact that $n_s$ is an empirical quantity, i.e., defined only by measurements, but assumed to be something that
physically exists. No study can know what $n_s$ for a particle is before conducting an ice nucleating experiment, thus it is empirically defined only.

- l. 42. Please change "The most..." to "One...".

- l. 44-45. Just because an experiment is inexpensive, easy, and yields many data points, does not make it a standard. Please state these as advantages instead.

- l. 45-47. This is inaccurate. The authors should be well aware of the countless cold stage experiments reproducing homogeneous freezing and expanding homogeneous freezing data sets without any contamination or surface effects. Please remove this sentence.

- l. 47-48. This is a bias sentence. According to Budke and Koop (2015) cross-contamination and evaporation was solved in Stopelli *et al.* (2014) using sealed tubes. If droplets or aliquot volumes are allowed to evaporate or to introduce contamination, then experiments were simple conducted wrong. Contrary to what the authors claim, results by (Budke and Koop, 2015) are not influence by these factors. This sentence must be removed.

- l. 49. There is no influence of the supporting surface. Again, previous studies reproduce homogeneous ice nucleation. See Thomas Koop, Ben Murray and Daniel Knopf groups. I will not do the authors literature search. I recommend the beginning of the sentence to change to the following. "We take a step further..."

- l. 63-64. Please remove the redundancy. I read in the previous paragraph and sentences that droplets are freely suspended.

- l. 65. What is the "nature" of a hydrometeor to the authors? It is not generated from a bulk solution and pipetted into position. Please remove this term.

- l. 249. It is redundant to say the experimental temperature is kept constant and the experiment is isothermal. Please remove this sentence.

- Figure 2, caption. Should the x-axis be labeled "$T_e$"? Should the inline equation also have $T_e$ instead of $T$?

- l. 274. The word "represent" is a little awkward here. I would reccomend the sentence to read "Freezing in M-WT experiments was observed under isothermal measurement conditions, and the stochastic approach was applied for data analysis."

- l. 279-280. Does this sentence really deserve its own paragraph?

- l. 350-351. In accord with the major comments. It should be stated here that if $S_p$ varies more than the authors expect, then a single component particle type may be erroneously identified as multiple component.

- l. 419-421. When sampling random error, the probability distribution from which numbers are sampled from should be stated. Did the authors sample from a normal distribution? According to the text and error bars in Fig. 5-7, the error is assumed to be normally distributed. The error in $n_s$ in Fig. 9, then is lognormally distributed? Did the authors sample frozen fractions from a normal distribution with or did they sample values of $n_s$? Please explain this in the text.

- l. 420-421. The authors assume the error bars are 95% confidence intervals. What data is this and how does that relate to the error bars on the previous graphs, which are all $1\sigma$ according to the captions? Of course, $1\sigma$ is not equivalent to 95% confidence. Their description is inadequate, and sounds like the authors sampled from some distribution, but threw away those values which were sampled beyond the 5 and 95% tail ends. In any case, the description and justification of their random sampling procedure needs to be explicit written.

- l. 416-418 and l. 422-423. This is redundant. Please rewrite.

- l. 423. Please change the text to read "...fitting a linear regression curve to the log-linear graph of randomly sampled data, and subsequently..."

- l. 431-434. The error on $\omega$ is washed over in this paragraph. The phrase, "$\omega$-based temperature shift", is first used here, however, the authors cannot expect a reader to formulate their own idea exactly how the shift and error on the shift is calculated by themselves. Please include around Eqns (12)-(14) details of to calculate these temperature shifts. This phrase "$\lambda$-based temperature shift" is used in the list of suggestions at the end of the manuscript, but I am uncertain what is being referred to. I would recommend to specifically define this terminology. Figure 10 is suppose to help with understanding this mathematical flow, however it only adds confusion because it has many undefined quantities that are not even included in the list of variables at the end of the manuscript. These unknown variables I found include $T_{cool}$, $n_{s,MAL}$, $\lambda = (0, 8)$, $T_{opt}$, $T_{\omega}$. Please state and explain the terminology, variables and equations used in this figure and include them in the list of variables at the end of the manuscript.

- l. 431-434. An addition question about this same paragraph. Is the random sampling of data also used to determine the error on $\omega$ in a similar way to $\lambda$? As of now, any equation or description of the error on $\omega$ is not clearly stated.

- l. 451. Do the authors mean deviations of the "simulated" data points?

- l. 537-538. There are only two locations in the manuscript where the authors use the phrase "cloud model", here and in the last line of the abstract. There is no discussion, argument or any information about cloud models in this paper. Therefore, no basis for any suggestion about cloud models is available. Please remove this suggestion, and remove the sentence about cloud models in the abstract.

- l. 541. I am sure the phrase "serve rather for orientation" means something specific to the authors, however this is not clearly defined in the manuscript. Would the authors please explain specifically what is meant by this?

- Figure B2. The labels and legend on the color scale are missing. Also, please state the simulated droplet size.

**References**

R. J. Herbert, B. J. Murray, T. F. Whale, S. J. Dobbie and J. D. Atkinson, *Atmos. Chem. Phys.*, 2014, **14**, 8501–8520.

G. Vali, *J. Atmos. Sci.*, 1994, **51**, 1843–1856.

P. A. Alpert and D. A. Knopf, *Atmos. Chem. Phys.*, 2016, **16**, 2083–2107.

D. A. Knopf, P. A. Alpert, A. Zipori, N. Reicher and Y. Rudich, *npj Clim. Atmos. Sci.*, 2020, **3**, 2.

S. Hartmann, H. Wex, T. Clauss, S. Augustin-Bauditz, D. Niedermeier, M. Rösch and F. Stratmann, *J. Atmos. Sci.*, 2016, **73**, 263–278.

C. Budke and T. Koop, *Atmos. Meas. Tech.*, 2015, **8**, 689–703.

E. Stopelli, F. Conen, L. Zimmermann, C. Alewell and C. E. Morris, *Atmos. Meas. Tech.*, 2014, **7**, 129–134.

---

## Referee Comment (RC2) · Anonymous Referee #2 · 5 Oct 2020

**1   Introduction**

In this work the authors use single droplet levitation techniques to analyze the freezing behavior ice nucleating particles (INP). Both isothermal and non-isothermal experiments are performed over a wide range of materials and conditions of atmospheric interest. The authors perform a detailed analysis of their experimental conditions, and of the ice nucleation behavior of the different materials. As extensive, single-droplet ice nucleation data are scarce, I find the data set is of interest to the atmospheric community. The analysis methods are however confusing and not appropriate for the type of experiments performed. They must be thoroughly clarified before the work could be published.

[Figure]

**2 Main Comments**

The analysis method defeats the purpose of carrying out single-droplet experiments. The authors interpret their results in terms of a previously developed model, appropriate for cold-stage type experiments, where the number of frozen events is counted out of a droplet population. It is certainly possible to relate the single-droplet and the population experiments (using a number of assumptions that must be clearly stated). But one wonders whether this is the best use of the single-droplet data. In the latter each of the analyzed freezing events is completely independent, and expressions like Eq.(1) (which is the same as Eq. 5) are not directly applicable. Instead one may expect that the statistics follow the usual Poisson distribution and be analyzed as such (the difference would be notorious when analyzing the non-isothermal experiments). Thinning of the distribution would help elucidate the presence of multiple components as well. Without attempting a more fundamental analysis, in terms of the actual statistics of single droplet events, it is hard to see what is new in this work.

The authors invest considerable effort to describe their experiments in terms of the active site density (INAS). However their results scream on a different direction: that the INAS approach is not appropriate to parameterize ice nucleation. Clearly time-dependency, particle-to-particle, and droplet-to-droplet variability must be accounted for. Looking at their results, one would expect the authors to call for a reanalysis of all of the ice nucleation data reported over the last decade. Instead, they go through considerable length trying to force the data into a flawed description of ice nucleation. This is a disservice to the scientific community and only works to perpetuate existing biases in the description of ice nucleation.

The suggestion that isothermal experiments should be analyzed with a time-dependent model whereas a time-independent model should be used in experiments with varying

temperature is wrong. If the authors are trying to elucidate the fundamental nature of ice nucleation, this should be independent of the analysis method. As mentioned above the time-independent formulation is at best a crude approximation hence and a time-dependent formulation should be emphasized.

There is nothing in the way the analysis is conducted that would suggest the existence of multiple components in the analyzed Kaolinite and sample. Fitting to a highly empirical model is not a sufficient condition, and given the results it suggests limitation of the assumed empirical model rather than a fundamental feature of the nucleation process. The authors suggestion that two distinct INP types would lead to a uneven distribution of nucleation efficiencies in the droplets (hence $\omega \neq \lambda$) is not supported by the isothermal experiments since this would also lead to two different slopes in Figure 6, and likely to a departure from the Poisson-type behavior.

It is also not clear what the authors mean by multiple component. The empirical correlations obtained in the Herbert et al. (2014) assume a normal distribution of the "b" parameter which amount to a collection of different nucleation sites. (hence different components?) The $\lambda$ parameter seems more related to the way the INP are dispersed in the droplet population hence it is just a feature of the experimental setup.

The authors state that they would make the data available upon request. This is appropriate during the review process. However to allow independent scrutiny the data supporting the plots must be placed on a permanent public repository before final publication.
**3  Other Comments**

- Line 27. This is not the nucleation rate.

- Line 32. Maybe another reason is that in the dry suspension method there is one particle per droplet, while in bulk measurements many particles are immersed within the same droplet. Hence there maybe slight differences in the environment around each active site.

- Line 90. The stochastic approach is also T-dependent. Please rephrase.

- Line 106. All sites must be equivalent as well.

- Line 127. I am having a hard time seeing any difference between this expression and Eq.(3). Is there anything here beyond semantics?

- Line 177. What is the basis to mix singular and time-dependent processes here? It would seem that they must be mutually exclusive.

- Line 186. This is not obvious at all. Please state clearly why this model is used, out of the many empirical approaches available.

- Line 189. Do you have to repeat the whole analysis if a different nominal cooling rate is used? Atmospheric cooling rates vary widely.

- Line 274. There is nothing preventing doing this analysis with the time-dependent model. In fact it would be a more rigorous approach.

- Line 311. It is not clear why binning of the results was required.

- Line 328. How do you go from INAS to the cumulative frozen spectra? What are the assumptions involved?

- Line 335. Please clarify this. Time is still involved even if you decide to ignore it.

- Line 345. Do these imply that the singular approach is invalid?

- Line 403. Still it seems that this would alter the temperature history of each active site.

- Line 409. I find this section very hard to follow. In fact I can't make sense of Figure 10. The authors go through a lot statistical dredging to try to fit the Herbert et al. (2014) model. The conclusions seem very dependent of the procedure used. There is hardly anything fundamental that can be extracted, especially not about multiple components.

- Line 465. If it is not applicable, why are the authors heavily using this model?

- Appendix. There is a lot of non well-behaved data that actually seems way more interesting than Kaolinite.

---

## Author Comment (AC1) · 18 Dec 2020

We thank both reviewers for their useful comments and suggestions that helped us to improve the manuscript.

We hereby reply to the questions and comments of Reviewer 1 in detail.

**Major issues**

I. First, there is insufficient review of previous studies that makes the manuscript unbalanced.

Descriptions of specific previous studies and their main claims using isothermal and cooling rate conditions are not included, but must be in order to be fair and balanced. Herbert et al. (2014) is extensively referenced for its data analysis, however it must be mentioned that they performed isothermal and cooling rate experiments on same particle types. Older studies such as Vali (1994) have cooling rate and isothermal conditions in a single experiment and support a multiple component approach held by the authors. Other studies, such as Alpert and Knopf (2016) and Knopf et al. (2020) come to a very different conclusions than the authors of the manuscript, but also used isothermal and cooling rate experiments. In summary, the literature review is incomplete and must be modified to include these references and any other relevant studies the authors wish to include.

Thank you for this note, we admit that the literature review was unbalanced. We oriented ourselves study to the widely used freezing assays that perform cooling rate experiments and apply the singular approach. In the revised version of the manuscript, we included also investigations where the authors applied stochastic models.

II. Additionally, assumptions about surface area and the impact of surface area calculations on their conclusions are not stated.

On I. 113, the authors state that in the case of interparticle variability, Eqn 3 cannot be used and Js is modified (Eqn 9) as a result. Eqn 3 also has surface area, Sp, which has its own uncertainty and variability. What is the estimated error on Sp and how does this error impact their findings? It is important the authors claim that the surface area in each droplet was not directly measured (only calculated from Eqn 4), and so the authors cannot rule out that droplet to droplet variability may be more than they expect. Along the same lines, the authors choose to vary cooling rate, however, varying surface area

could also be done to test for single or multiple components. For example, Hartmann et al.

(2016) changed particle mobility diameter from 0.3- 0.7 m and found similar ns values for kaolinite. It would follow that kaolinite would be considered as a single component particle type under sizes relevant for the atmosphere, in contrast to this study in which the authors labeled kaolinite as multiple component. In light of the work of Hartmann et al. (2016), I would ask the authors to include some discussion about changing particle size and the impact on what could be declared as multiple or single component. The authors should state the upper size limit of single particles in their droplets for each type.

The actual concentration of particles in the droplets was not measured. The particle surface areas used for the calculations in the paper were calculated from the particle concentration in the droplet, the droplet volume, and the specific surface area of the INP. We considered all measurement error sources for calculating the propagated error. In order to reduce experimental uncertainty, a homogeneous solution was generated and used for droplet generation. Although efforts were made to unify and standardize the sample generation (also following the suggestions of Hiranuma et al., 2018), we cannot rule out INP surface area variation among the investigated droplets. That can significantly influence the nucleation description (Alpert and Knopf, 2016). There are several sources of error which might increase the surface area uncertainty, like: externally or internally mixed particles, size distribution of the particles, aggregation due to sedimentation and internal circulation. The most appropriate way would be the continuous measurement of the surface area inside each droplet under investigation, but that seems not feasible currently. Furthermore, the ice active site densities may vary on microscopically identical (i.e. size, chemical composition) particles.

We varied the total surface inside the droplets by using different particle concentrations in aqueous solutions. Such experiments resulted in consistent  $n_s$  values (see Fig. 5.). Unfortunately, we cannot provide size limits of the particles we used. We used bulk particle samples and in a relatively high concentration. Furthermore, aggregation in an aqueous solution would anyway modify the dry particle size distribution. We discussed these points in the revised manuscript.

III. There is a fundamental flaw with shifting observed temperatures outside their investigated range.

Applying a temperature shift is fundamentally flawed. For example, the warmest observed temperature that feldspar nucleates in this manuscript is about 263 K. However, the temperature correction is made to 269, which is 5 K warmer than measured. The authors did not measure nucleation at this warmer temperature. This is extrapolating data outside of the observed temperature range. Why don't the authors shift ns instead? Afterall, temperature is measured. ns is not measured, but it is calculated from measurements. This is a caveat that must be claimed and the reader must be warned about using data outside of the authors measured temperature range.

The temperature was shifted to higher values but remained still within the investigated range. The droplet injected into M-AL was adaptively cooling from some positive degrees to below -25 °C. The surface temperature was continuously measured by means of an infrared thermometer. Therefore, all freezing events that occurred at temperatures between 0° and -25 °C were captured. In the case

of the M-WT measurements, the necessary temperature shift was small and within the wind tunnel air temperature variation.

We observed a systematic offset of our data points in intercomparison campaigns (INUIT and FINO2). For some particle types this offset was obvious, and for some it was negligible, i.e. within the measurement error. By seeking for the reason of this offset, first we checked the calculated ns values and possible errors in the calculation or in the sample treatment. However, since the offset appeared for different concentrations, and also when treating the samples following the experimental protocols of the campaigns, we investigated the shift in the temperature. Finally, we made measurements on the internal temperature of the droplets, before we arrived to the temperature shift caused by the change in cooling rate.

IV. Finally, it must be acknowledged, that their conclusions about a temperature shift or a particle type being single or multiple component is dependent entirely on their choice of data analysis procedure and not a fundamental property of the types of particles.

Another major concern is that the formulation here and in Herbert et al. (2014) is purely empirical and not based on any physical theory. Therefore, a particle type declared by the authors as a single and multiple component, or having a material dependent temperature correction factor (shift) is also empirical and not a fundamental property. This is a necessary caveat that must be stated. I do not wish to discount the evidence the authors give for either, however, they do need to stress that there is no direct observation of single or multiple components.

We corrected the formulation regarding the classification of particles as single or multiplecomponent. Instead, we stressed that the freezing behaviour of the particles can be described by a single or a multiple-component approach. Furthermore, we did not aim to develop a new model or framework for immersion freezing, but rather to provide new experimental data to check whether they can be described by the existing approaches, and how the results obtained from two experimental techniques match. The main motivation of the study was the freezing temperature shift observed in our M-AL measurements during intercomparison campaigns (INUIT and FINO2). The deviations often visible in intercomparisons from different experimental techniques are still not clarified. We believe that we can support other experimentalists facing with similar problems, and probably attract attention for this very important issue of freezing temperature shift.

**Minor issues**

1. There are frequent typos and instances of comma misuse. In addition, there are multiple instances of greek letters being spelled out instead of actually using the symbol. This lack of proofreading is not appropriate for a manuscript submission and shows a lack of care for their own work. I urge the authors to correct this.

We thoroughly reread the manuscript and corrected some typos and erroneously written instances of mathematical and Greek symbols. Although we appreciate the reviewer's opinion but we do not agree with the note on the lack of our proofreading.

 I. 11-12. In the abstract, the authors write the words "material dependent correction factor". However, it is not claimed what is incorrect. That would be helpful in the abstract. The measured temperature is not incorrectly measured, so please be precise and tell the reader what is really being corrected.

We modified the sentence following the suggestion.

3. I. 20. Please change "The nucleation abilities. . . " to "The ice nucleation abilities... "

Corrected.

4. I. 25-27. This is an inaccurate sentence. ns is calculated from the total number of nucleation events per unit surface area of the particles, and then it is assumed to be equivalent to the number of sites on the particles. This is a big difference to what is written. The point here is to stress the fact that ns is an empirical quantity, i.e., defined only by measurements, but assumed to be something that physically exists. No study can know what ns for a particle is before conducting an ice nucleating experiment, thus it is empirically defined only.

The sentence was corrected following the reviewer's suggestion, and now reads as "This is calculated from the experimentally determined total number of nucleation events per unit surface area of the particles. INAS density is used to represent the number of ice active sites on the particles that are active between 0 °C and the sub-zero temperature "

5. I. 42. Please change "The most. . . " to "One. . . ".

Done.

6. l. 44-45. Just because an experiment is inexpensive, easy, and yields many data points, does not make it a standard. Please state these as advantages instead.

We modified the sentence as follows: "Their advantages of inexpensive and easy operation, and the large number of simultaneously measurable droplets offering good count statistics, promoted them for INP characterization experiments."

7. I. 45-47. This is inaccurate. The authors should be well aware of the countless cold stage experiments reproducing homogeneous freezing and expanding homogeneous freezing data sets without any contamination or surface effects. Please remove this sentence.

The sentence has been removed.

I. 47-48. This is a bias sentence. According to Budke and Koop (2015) crosscontamination and evaporation was solved in Stopelli et al. (2014) using sealed tubes. If droplets or aliquot volumes are allowed to evaporate or to introduce contamination, then experiments were simple conducted wrong. Contrary to what the authors claim, results by (Budke and Koop, 2015) are not influence by these factors. This sentence must be removed.

The sentence has been removed.

9. I. 49. There is no influence of the supporting surface. Again, previous studies reproduce homogeneous ice nucleation. See Thomas Koop, Ben Murray and Daniel Knopf groups. I will not do the authors literature search. I recommend the beginning of the sentence to change to the following. "We take a step further..."

We modified the sentence to the following: "In our study we take a step further to real atmospheric conditions of cloud droplets, and avoid the contact of any supporting surface. The single droplet levitation techniques employed offer experiments ..."

10. l. 63-64. Please remove the redundancy. I read in the previous paragraph and sentences that droplets are freely suspended.

Done.

11. I. 65. What is the "nature" of a hydrometeor to the authors? It is not generated from a bulk solution and pipetted into position. Please remove this term.

We removed the whole sentence.

12. I. 249. It is redundant to say the experimental temperature is kept constant and the experiment is isothermal. Please remove this sentence.

The sentence has been deleted.

13. Figure 2, caption. Should the x-axis be labelled "Te"? Should the inline equation also have Te instead of T?

The figure shows the results of the calculation of the approaching time of the surface temperature to equilibrium at different air temperatures. We corrected the figure caption, the axis label and the inline equation accordingly.

14. I. 274. The word "represent" is a little awkward here. I would recommend the sentence to read "Freezing in M-WT experiments was observed under isothermal measurement conditions, and the stochastic approach was applied for data analysis."

We modified the sentence according to the reviewer's suggestion as, "Immersion freezing in M-WT experiments was investigated under isothermal measurement conditions, hence, the stochastic approach was applied for data analysis."

15. l. 279-280. Does this sentence really deserve its own paragraph?

Corrected.

16. l. 350-351. In accord with the major comments. It should be stated here that if Sp varies more than the authors expect, then a single component particle type may be erroneously identified as multiple component.

We carried out careful analysis and considered the measurement uncertainty in order to classify the particle types as correct as possible, but we agree with the reviewer, that the estimated total surface area may vary significantly more than we expected. Therefore, we added the following sentences to this discussion: "In our experiments the total surface area A was estimated from the concentration of the aqueous solution and from the specific surface area. To accurately measure the actual total surface area of INP inside the droplets, which should be taken into account for calculating  $\omega$  and  $J_s$ , is currently not feasible. Therefore, the error of A might be significantly higher than estimated, which would result in a false classification of the INP as single-component."

17. I. 419-421. When sampling random error, the probability distribution from which numbers are sampled from should be stated. Did the authors sample from a normal distribution? According to the text and error bars in Fig. 5-7, the error is assumed to be normally distributed. The error in ns in Fig. 9, then is lognormally distributed? Did the authors sample frozen fractions from a normal distribution with or did they sample values of ns? Please explain this in the text.

We sampled values from  $n_s$ . We did not consider the distribution of  $n_s$  but randomly took  $n_s$  values falling within the 1 $\sigma$  bounds around the mean  $n_s$  value. This might overestimate the  $\lambda$  error.

18. I. 420-421. The authors assume the error bars are 95% confidence intervals. What data is this and how does that relate to the error bars on the previous graphs, which are all 1σ according to the captions? Of course, 1σ is not equivalent to 95% confidence. Their description is inadequate, and sounds like the authors sampled from some distribution, but threw away those values which were sampled beyond the 5 and 95% tail ends. In any case, the description and justification of their random sampling procedure needs to be explicit written.

Thank you for the note. We used here also the  $1\sigma$  errors, not the 95% confidence intervals. We corrected the text, and we reformulated the description of the procedure to estimate the error of  $\lambda$ .

19. l. 416-418 and l. 422-423. This is redundant. Please rewrite.

We deleted the unnecessary part of the sentence in line 423.

20. I. 423. Please change the text to read "... fitting a linear regression curve to the log-linear graph of randomly sampled data, and subsequently..."

This part of the sentence was deleted (s. our reply to the last comment).

21. I. 431-434. The error on  $\omega$  is washed over in this paragraph. The phrase, " $\omega$ -based temperature shift", is first used here, however, the authors cannot expect a reader to formulate their own idea exactly how the shift and error on the shift is calculated by themselves. Please include around Eqns (12)-(14) details of to calculate these temperature shifts. This phrase " $\lambda$ -based temperature shift" is used in the list of suggestions at the end of the manuscript, but I am uncertain what is being referred to. I would recommend to specifically define this terminology. Figure 10 is suppose to help with understanding this mathematical flow, however it only adds confusion because it has many undefined quantities that are not even included in the list of variables at the end of the manuscript. These unknown variables I found include Tcool, ns,MAL,  $\lambda$ = (0, 8), Topt, T $\omega$ . Please state and explain the terminology, variables and equations used in this figure and include them in the list of variables at the end of the manuscript.

The phrase " $\omega$ -based temperature shift" has been removed, and the sentence has been reformulated to explicitly refer to the equation used here.

We decided to move Figure 10 to the Appendix. Our intention was to depict the procedure and help the reader to understand the process, but apparently it confused both reviewers. We added the variables to the list of variables.

22. l. 431-434. An addition question about this same paragraph. Is the random sampling of data also used to determine the error on  $\omega$  in a similar way to  $\lambda$ ? As of now, any equation or description of the error on  $\omega$  is not clearly stated.

The error of  $\omega$  is the standard error of the linear fit on the ln(R/A) vs. T curve. This is now explicitly written in the text.

23. I. 451. Do the authors mean deviations of the "simulated" data points?

Here the corrected data points, i.e. when shifted to higher temperatures are meant. The sentence was rewritten.

24. I. 537-538. There are only two locations in the manuscript where the authors use the phrase "cloud model", here and in the last line of the abstract. There is no discussion, argument or any information about cloud models in this paper. Therefore, no basis for any suggestion about cloud models is available. Please remove this suggestion, and remove the sentence about cloud models in the abstract.

We removed the sentence about cloud models in the abstract. Nevertheless, in our opinion the freezing temperature shift stemming from the change in the cooling rate is relevant for cloud modelers and is, therefore, presented as a suggestion in Conclusions and Suggestions.

25. l. 541. I am sure the phrase "serve rather for orientation" means something specific to the authors, however this is not clearly defined in the manuscript. Would the authors please explain specifically what is meant by this?

What is meant here is that the apparent temperature shift depends on the specific aerosol sample that is investigated. We want to avoid that any reader would use the temperature shifts tabulated in our paper or in Herbert et al. (2014) "as it is", because it might vary due to the chemical composition of the sample. Rather, in our opinion cooling experiments should not be conducted using exclusively one cooling rate or without the simultaneous measurement of the nucleation rate coefficient.

26. Figure B2. The labels and legend on the color scale are missing. Also, please state the simulated droplet size.

The figure was corrected by including the labels and the legend, and the drop size is now indicated in the figure caption.

---

## Author Comment (AC2)

We thank both reviewers for their useful comments and suggestions that helped us to improve the manuscript.

We hereby reply to the questions and comments of Reviewer 2 in detail.

**Major comments**

I. The authors interpret their results in terms of a previously developed model, appropriate for cold-stage type experiments, where the number of frozen events is counted out of a droplet population. It is certainly possible to relate the single-droplet and the population experiments (using a number of assumptions that must be clearly stated). But one wonders whether this is the best use of the single-droplet data. In the latter each of the analyzed freezing events is completely independent, and expressions like Eq.(1) (which is the same as Eq. 5) are not directly applicable. Instead one may expect that the statistics follow the usual Poisson distribution and be analyzed as such (the difference would be notorious when analyzing the non-isothermal experiments).

One has to distinguish between the two single-droplet levitation techniques we used. The isothermal (M-WT) measurements can be described by the stochastic approach. For the cooling experiments of M-AL the singular approach, which is also used for cold stage array experiments, can be applied. The droplet freezing in cold stage experiments is usually also considered as completely independent, and Eq. 1 is applied. In our experiments we use the same samples, generate the drops in a similar way as in clod stage experiments. Each drop in M-Al are cooled down similarly, although not identically. But that might be the case on a cold stage, where temperature differences on the surface can result in different drop temperatures. We definitely relate one drop temperature to another. In this way, M-AL and cold stage measurement are comparable in terms of  $f_{ice}$  and  $n_s$ .

II. Thinning of the distribution would help elucidate the presence of multiple components as well.

There are many possible sources for the presence of multiple components. One important is the size distribution. Other sources are the internal and external mixture of the sample material, or any physical, chemical, or biological contamination. In summary, in an ideal experimental case, a very pure INP material with thin size distribution would be immersed in a droplet under investigation. Furthermore, the total surface area is a crucial parameter. In order to use the correct value of the total surface area in the calculation, it should be measured directly inside the droplets. That would help avoiding error originating from aggregation in aqueous solutions, for example. Unfortunately, our experimental setup was not sufficient to carry out such measurements. Instead, our instrumentation allows the investigation of the freezing process under conditions that are more realistic, like the free levitation in an airflow, for instance. We believe that our measurements can help researchers making predictions for real atmospheric processes when utilizing results from, e.g., cold stage experiments.

III. The authors invest considerable effort to describe their experiments in terms of the active site density (INAS). However their results scream on a different direction: that the INAS approach is not appropriate to parameterize ice nucleation. Clearly time-dependency, particle-to-particle, and droplet-to-droplet variability must be accounted for. Looking at their results, one would expect the authors to call for a reanalysis of all of the ice nucleation data reported over the last decade. Instead, they go through considerable length trying to force the data into a flawed description of ice nucleation. This is a disservice to the scientific community and only works to perpetuate existing biases in the description of ice nucleation.

INAS is a concept emerging from experiments and broadened in the community due to its easy implementation into cloud models. Our aim was to contribute to the justification or disproof of its usage in applications describing cloud processes. Our setup simulates the cloud conditions in a much appropriate way concerning the flow, shape and contact-free levitation. Therefore, we converted our measured data to INAS density. Nonetheless, we also provide results of the heterogeneous nucleation rate coefficients from our measurements under isothermal conditions.

IV. The suggestion that isothermal experiments should be analyzed with a time-dependent model whereas a time-independent model should be used in experiments with varying temperature is wrong. If the authors are trying to elucidate the fundamental nature of ice nucleation, this should be independent of the analysis method. As mentioned above the time-independent formulation is at best a crude approximation hence and a time-dependent formulation should be emphasized.

Ice nucleation is a stochastic, i.e. time dependent process. The stochastic approach is based on classical nucleation theory and represents a physical description. The singular description is an empirical approach, which was introduced to explain ice nucleation in a simplified manner. The temperature dependence is neglected as it is assumed that critical clusters form on ice-active sites at characteristic temperatures. The singular approach has been used to compare the results from different experimental techniques via the ns values (e.g., Hiranuma et al., 2015; Wex et al., 2015) Furthermore, the singular description can be easily implemented in cloud models (e.g. Diehl et al, 2015, ACP). If the  $\lambda$  value is small, a large temperature shift is predicted in accord with the stochastic model (s. Theoretical background in the revised version of our manuscript). In the contrary, large  $\lambda$  values result in small temperature shifts, which are in most cases negligible at least in terms of the measurement uncertainties. In these cases, the application of the singular approach is rather justified. Thus, our study helps to elucidate the limitations of the singular approach used by dozens of experimental studies by providing experimental data of  $\lambda$  for a set of INP. We critically revised our paper to avoid any misleading formulation regarding this topic.

V. There is nothing in the way the analysis is conducted that would suggest the existence of multiple components in the analyzed Kaolinite and sample. Fitting to a highly empirical model is not a sufficient condition, and given the results it suggests limitation of the assumed empirical model rather than a fundamental feature of the nucleation process. The authors suggestion that two distinct INP types would lead to a uneven distribution of nucleation efficiencies in the droplets (hence  $\omega \neq \lambda$ ) is not supported by the isothermal experiments since this would also lead to two different slopes in Figure 6, and likely to a departure from the Poisson-type behavior.

The formulation in the manuscript was probably misleading. We used the term multiple component because Herbert et al. applied this approach. In this context, multiple component may mean internally or externally mixed particles, contaminations, etc., but also the high scatter of contact angles, or the INP surface area variability in the individual droplets. Hence, the effect of particle variability is more important than the time dependence of nucleation. For single component systems the stochastic model has to be applied, whereas for multiple components the singular approach may also be valid. By revision of the manuscript we payed attention to address this issue.

VI. It is also not clear what the authors mean by multiple component. The empirical correlations obtained in the Herbert et al. (2014) assume a normal distribution of the "b" parameter which amount to a collection of different nucleation sites. (hence different components?) The parameter seems more related to the way the INP are dispersed in the droplet population hence it is just a feature of the experimental setup.

Please see our reply to the last comment. Of course we cannot completely rule out the effect of our sample preparation or experimental features. That was one reason why we described our procedures in details, and carried out statistical tests on analysis results. Future studies may clarify this.

VII. The authors state that they would make the data available upon request. This is appropriate during the review process. However to allow independent scrutiny the data supporting the plots must be placed on a permanent public repository before final publication.

We are not sure what the reviewer's request is. We can certainly publish the data points and errors shown in the figures before publication, in case the manuscript will be accepted. Publication of raw experimental data on a permanent public repository will follow after final publication.

**Minor comments**

1. Line 27. This is not the nucleation rate.

Yes, thank you, we corrected it into the nucleation rate coefficient.

 Line 32. Maybe another reason is that in the dry suspension method there is one particle per droplet, while in bulk measurements many particles are immersed within the same droplet. Hence there maybe slight differences in the environment around each active site.

Yes, we absolutely agree. Since this issue is well beyond the scope of the present paper, and because we do not possess the instrumental possibility to adequately study this discrepancy between dry dispersion and aqueous suspension techniques, we did not speculate on the reason for it. The main message we want to pass over here is how important the rigorous examination of the limitations of one's measurement technique is (which should actually be evident, but in fact is often not the case). Furthermore, this part served for orientation for the reader that the paper deals with only one type of immersion freezing measurement methods, namely the aqueous suspension technique.

3. Line 90. The stochastic approach is also T-dependent. Please rephrase.

The temperature dependency of the stochastic approach is indicated.

4. Line 106. All sites must be equivalent as well.

Yes, thank you, we corrected the sentence.

5. Line 127. I am having a hard time seeing any difference between this expression and Eq.(3). Is there anything here beyond semantics?

In Eq. (5) time dependency is not included (singular approach), while in Eq. (3) both time and temperature dependency are involved (stochastic approach).

6. Line 177. What is the basis to mix singular and time-dependent processes here? It would seem that they must be mutually exclusive.

We reconstructed the entire section on the theoretical background, and reformulated the approaches in a more consistent and clear way.

7. Line 186. This is not obvious at all. Please state clearly why this model is used, out of the many empirical approaches available.

Please see our reply on the last comment.

8. Line 189. Do you have to repeat the whole analysis if a different nominal cooling rate is used? Atmospheric cooling rates vary widely.

Yes, a cooling rate differing from 1 K/min would cause a different freezing temperature shift. This has to be counted for when comparing measuring devices, but also has to be taken into account in cloud models.

9. Line 274. There is nothing preventing doing this analysis with the time-dependent model. In fact it would be a more rigorous approach.

The primary goal of the current study was the investigation of the temperature shift in cooling rate experiments. The isothermal conditions in M-WT represented the physical basis allowing to interpret the experimental results using the time-dependent model. The M-AL measurements provide INAS densities utilizing the singular approach. Since this instrument exhibits high and varying cooling rates, the implementation of the time-dependent model was first abandoned and we restricted our analysis to the singular approach.

10. Line 311. It is not clear why binning of the results was required.

Binning means in M-AL experiments the counts of individual drops frozen in temperature intervals between T-0.5 K and T+0.5K. This was necessary because the measurement setups has a temperature uncertainty of +/- 0.5 K.

11. Line 328. How do you go from INAS to the cumulative frozen spectra? What are the assumptions involved?

The sentence was corrected, and now reads "Figure 5 shows the INAS densities computed using Eq. (5) from  $f_{ice}$  spectra obtained from M-WT measurements of kaolinite (...)"

12. Line 335. Please clarify this. Time is still involved even if you decide to ignore it.

Yes, time is involved, however, using fixed times of 30 s in M-WT experiments we expressed  $n_s$  in terms of  $f_{ice}$ . Herbert provided an equation for calculating the time an isothermal experiment needs to reach the same frozen fraction as a cooling rate experiment (Eq. 19 in Herbert et al., 2014):

$$t = \frac{1}{\lambda \cdot r}$$

with r being the cooling rate. Assuming  $\lambda$ =2 and using the standard cooling rate of 1 K/min, t = 30s. In this approach, the stochastic element is considered to represent the random occurrence of an ice nucleating site somewhere on the INP surface (see Vali, 2014, Eqs. 12 and 13).

13. Line 345. Do these imply that the singular approach is invalid?

The droplet freezing in the M-WT can be described by the stochastic approach. In order to compare cooling rate and isothermal experiments, we took the accumulated data for the total observation time of 30 seconds of the isothermal measurements into account for calculating  $f_{ice}$ .

14. Line 403. Still it seems that this would alter the temperature history of each active site.

Yes, we agree. As described in the manuscript, we believe that the huge amount of kaolinite particles, and hence, of the active sites, is so large that the warmer temperature in the drop interior does not play any role in freezing initiation. Furthermore, the temperature history lasts not more than 80 seconds in M-AL experiments, which might be too short for significantly affecting the nucleation ability of active sites.

15. Line 409. I find this section very hard to follow. In fact I can't make sense of Figure 10. The authors go through a lot statistical dredging to try to fit the Herbert et al. (2014) model. The conclusions seem very dependent of the procedure used. There is hardly anything fundamental that can be extracted, especially not about multiple components.

We decided to move Figure 10 to the Appendix. Our intention was to depict the procedure and help the reader to understand the process, but apparently it confused both reviewers. We added the variables to the list of variables.

The aim of this section was to introduce the procedure for calculating  $\lambda$ , and to estimate whether it differs from  $\omega$ . This implies whether the INP can be described by single component stochastic approach or not. Unlike Herbert et al., we investigated whether  $\lambda$  equals  $\omega$  in terms of our measurement uncertainties. In this regard, the conclusions depend on the procedure and instrumentation used.

**16. Line 465. If it is not applicable, why are the authors heavily using this model?**

The Herbert et al. approach served as basis for our analysis. Since Herbert et al. used constant cooling, their concept had to be modified and adapted to our experiments in which the cooling rate was varying. In addition, we improved the analysis method by introducing the procedure considering data scatter and measurement errors.

We found some noticeable behaviour of our INAS density results from M-AL immersion freezing measurements when compared to other techniques and devices within the INUIT and FINO2 campaigns. We could observe an apparent temperature shift in our results, the shift being different for different materials, and freezing temperatures. We were trying to figure out whether it was a measurement artefact or a phenomenon with a physical basis. This is how we came to the Herbert approach, which adequately modelled our experimental findings.

17. Appendix. There is a lot of non well-behaved data that actually seems way more interesting than Kaolinite.

We chose kaolinite to demonstrate the procedure we used to analyse our experimental data. Kaolinite was also analysed in several immersion freezing studies, as in Herbert et al., for instance. Nevertheless, we agree that there are other interesting data in our dataset, and we are happy that the reviewer agrees with that. We are going to share the measurement data in a scientific repository for other researchers, as mentioned earlier.

---

## Author Response (AR2)

Dear Prof. Knopf,

Thank you for your revision of our manuscript and for your comments. Here we provide our replies to them:

1. Throughout the manuscript when discussing lambda and omega values the units are missing. You indicate both parameters are in units "K-1". However, for a reader not familiar with this approach, Eq. (10) does not result in a parameter with this unit, but units of omega appear to be ln(s-1 cm-2)/K. This likely reflects the empirical nature of this approach (maybe an approximation). Lambda is also a proportionality factor and its units are correct. For the purpose discussing these two proportionality factors, I am fine as is in this manuscript, however, you need to elaborate on the units of omega (especially since it is different from Eq. 7 by surface area from Vali 2014), so a reader going through the math understands the approach of this method and how you can compare it to lambda.

The units of lambda and omega indeed appear to be ln(s-1 cm-2)/K. This stems from their similar definition in equations 7 and 10, and that the units of R/A and J are the same. Nevertheless, the quantity with ln is generally (but incorrectly) treated as dimensionless in order to get rid of this awkward unit above (e.g., in Herbert et al, 2014; or Budke and Koop, 2015). This reflects the empirical definition of lambda and omega. We added these information to the discussion after equations 7 and 10.

2. Line 28: Typo "coefficient".

Corrected.

3. Line 204, after Eq. 15: I believe there is an error in derived nucleation time. Should it be "....1 K min-1, and a typical lambda value of 2 K-1, a total observation time of 30 seconds is obtained."?

Yes, thank you for this catch. We have corrected to "30 seconds"

4. Line 274: Please include newest references on this matter. Please add reference Knopf et al. (2020) to Alpert and Knopf (2016) that provides proof that surface area variance and stochasticity explains NX illite immersion freezing, also topic of this study. The issue of surface variability was also discussed in Barahona (2020) and could be mentioned in separate sentence. Knopf et al. (2020) and Barahona (2020) should be included in conclusions section on page 26.

We extended the discussion near line 274 and the conclusion with the findings of Knopf et al. (2020) and Barahona (2020).

Furthermore, we submitted the data sets plotted on the figures to zenodo.org. They will be published and available under doi:10.5281/zenodo.4436153 as given now in line 606. The final publishing of the data set will be conducted as soon as the doi of the published paper is generated.

Yours sincerely,

Miklós Szakáll

Institute for Atmospheric Physics Physik der Atmosphäre Johannes Gutenberg University Mainz, Germany